# FLIP: Towards Comprehensive and Reliable Evaluation of Federated Prompt Learning

**Dongping Liao**
State Key Lab of IoTSC,
CIS Dept, University of Macau.
yb97428@um.edu.mo

**Xitong Gao**[*]
SIAT, CAS.
SUAT.
xt.gao@siat.ac.cn

**Chengzhong Xu**[*]
State Key Lab of IoTSC,
CIS Dept, University of Macau.
czxu@um.edu.mo

## Abstract

The increasing emphasis on privacy and data security has driven the adoption of federated learning (FL). Prompt learning (PL), which fine-tunes prompt embeddings of pretrained models, has gained a surge of interest in FL community, marked by the emergence of an influx of federated prompt learning (FPL) algorithms. Despite recent advancements, a systematic understanding of their underlying mechanisms and principled guidelines for deploying these techniques in different FL scenarios remain absent. Moreover, inconsistent experimental protocols, limited evaluation scenarios, and the lack of the proper assessment of centralized PL methods in existing works have obscured the essence of these algorithms. To close these gaps, we introduce a comprehensive benchmark, named FLIP, to achieve standardized FPL evaluation. FLIP assesses the performance of 13 centralized and FPL methods across 3 FL protocols and 12 open datasets, considering 6 distinct evaluation scenarios. Our findings demonstrate that PL maintains strong generalization performance in both in-distribution and out-of-distribution settings with minimal resource consumption, but there is no silver bullet found for diverse FPL scenarios. The results (1) pinpoint the suitable application scenarios of each FPL algorithm, (2) demonstrate the competitiveness of adapted centralized PL methods, and (3) offer notable insights to interpret their effectiveness and remaining challenges. All benchmarks and code are available to facilitate further research in this domain[2].

## 1 Introduction

User awareness of privacy and data security and recent legislation such as the General Data Protection Regulation (GDPR) [1, 2, 3] have led to the recent rise of federated learning (FL) [4, 5, 6, 7, 8] as a promising approach to training machine learning models without the need to share the data itself. Despite the potential benefits of FL, it introduces large computational and communication overheads due to the need to synchronize models training and data heterogeneity across devices.

Leveraging large-scale text-image aligned data, pretrained vision-language models such as CLIP [9] and ALIGN [10] show strong zero-shot image classification performance. Instead of training new models, *prompt learning* (PL) [11, 12, 13, 14, 15, 16] fine-tunes the prompt embeddings of these pretrained vision-language models. This technique has demonstrated strong performance, requiring only one or two examples per class in the in-distribution setting, and also excels in domain generalization.

In the context of FL, the advantages of federated prompt learning (FPL) are: (1) it incurs lower computational costs compared to training a model from scratch; (2) it significantly reduces com-

---

[*]Corresponding authors.

[2]The codebase and full benchmark suite are available at https://github.com/0-ml/flip.

munication overheads and memory requirements, as the trainable prompt embeddings are much smaller than the model's weights; (3) properly utilizing the pretrained model's knowledge can gain strong in-distribution and out-of-distribution generalization performance. However, in contrast to the centralized PL settings, we envision that FL brings new perspectives specific to the paradigm:

- **How effective are global and personalized prompt models?** Inter-device data heterogeneity can naturally arise, potentially leading to slower convergence and degraded generalization performance of prompts. It remains to be seen how PL algorithms can train shared global prompts that effectively adapt to data heterogeneity. Conversely, personalized FL trains client-specific models to address conflicting distributional shifts across devices that arise due to data heterogeneity. Considering the strong generalization performance of PL, it is intriguing to explore how it may learn effective personalized prompts.

- **Impact of various data distribution shifts on FPL.** Prompt learning algorithms show robust generalization capabilities across diverse distributional shift scenarios [12, 13, 17]. This characteristic is particularly beneficial within federated environments, where devices often operate under constraints of limited data and computational resources. Our objective is to assess the effectiveness of FPL algorithms in scenarios characterized by the following: **data scarcity** (few-shot learning), **unseen classes** (test data containing classes not present during training), and **cross-domain** distributional shifts (test data comprising the same classes as training data but from different domains).

- **Cost-effectiveness of FPL.** Prompt learning in federated settings introduces hyperparameters, such as the number of prompts and prompt length, reflecting the trade-off between model performance and communication cost. In addition, different baselines may exhibit varying convergence rates and have different per round computational and communication costs. Our goal is to conduct sensitivity analyses of various algorithms to examine how different setups may influence their respective trade-off relationship and algorithm choices.

Beyond providing insights to these questions, we also aim to deliver a comprehensive and extensible benchmark of the FPL baselines, along with a suite of evaluation metrics that rigorously measure algorithmic performance. To quantify the effectiveness of FPL algorithms and to provide a standardized benchmark for rapid, reproducible, and reliable evaluations in FPL, we make the following contributions:

- We designed and implemented FLIP, a unified, modular and open-source codebase with unified training and evaluation procedures and interface, comprising a suite of vision-language models, datasets, faithful implementations of algorithms, and evaluation metrics. For ease of use, it has swappable modules for PL algorithms and FL strategies.

- We systematically evaluated the performance of FPL algorithms under various challenging scenarios, including global and personalized learning under data heterogeneity, few-shot, novel-class, and cross-domain distributional shifts. Our benchmark also includes a rich set of methods adapted from centralized PL, which are generally overlooked in the evaluation of existing FPL papers. Moreover, we explored the trade-off relationship between model performance and communication cost under different PL configurations.

- Finally, FLIP provides a standardized comprehensive benchmark suite and we carried out the extensive experiments for the 13 FPL baseline algorithms under 3 FL protocols, with 6 metric-reporting scenarios on 12 open datasets. The FLIP codebase is fully open-source and publicly available for our community.

## 2 Related Work

**Prompt learning of vision-language pretrained models.** The recent advent of vision-language models (VLMs), such as CLIP [9] and ALIGN [10], which learn to align text and image pairs in a shared embedding space using contrastive learning, marks a significant milestone in vision-language understanding, showing remarkable zero-shot performance on various downstream tasks. Leveraging the pretrained VLMs, CoOp [11] introduces the optimization of learnable text context embeddings to adapt pretrained VLMs to improve downstream performance. CoCoOp [12] extends CoOp by further training a Meta-Net to predict suitable prompt embeddings from image features. To prevent PL from converging to a single point, Prompt Learning with Optimal Transport (PLOT) [18] formulates

PL as optimal transport between the cosine distances of visual features and the prompt features. Prompt Distribution Learning (ProDA) [13] optimizes text prompts by learning to model Gaussian distributions over the prompt embeddings, and encourages semantic orthogonality among the prompt embedding vectors to enhance the generalization capability. On a similar note, Prompt-aligned gradient (ProGrad) [19] aligns the prompt gradients that are in conflicting directions with the zero-shot predictions, while self-regulating prompts (SRC) [17] propose to condition prompted features to be consistent with the CLIP features with self-consistency regularization, and knowledge-guided contextual optimization (KgCoOp) [15] regularizes the prompt embeddings to be within the proximity of hand-crafted prompts.

**FPL for vision-language model adaptation.** Early endeavor in fine-tuning pretrained vision-language models in a federated setting has been explored in [20] to address the challenges of inter-client data heterogeneity and improve generalization performance under cross-domain scenarios. Doing so, however, incurs a heavy communication cost of full model weights. Instead, PL algorithms in the federated setting can significantly reduce communication overheads by only transmitting the prompt embeddings, or a small network that predicts the prompt embeddings. PromptFL [21] extends CoOp [11] to the federated setting for the server aggregation of prompt embeddings using FedAvg [4]. To mitigate feature and label shifts, FedOTP [16] introduces a strategy where a shared global prompt is learned to extract consensus information, along with a local prompt for each client to capture client-specific knowledge. Subsequently, it applies an optimal transport-based alignment to regularize both the global and local prompts, balancing global consensus and local personalization. Following a similar direction, PromptFolio [22] borrows concepts from portfolio optimization to manage a collection of diverse global and local prompts. FedPGP [23], addresses the global-local trade-off through low-rank adaptation techniques complemented by contrastive learning objectives. FedTPG [24] leverages a cross-attention module to generate prompts conditioned on task-related text input. pFedPG [25] and SGPT [26] explores prompt adaptation and selection for personalized FPL. FPL has also been investigated under the domain generalization problem [27, 28]. DP-FPL [29] introduces differential privacy to enhance the privacy protection of FPL.

**FL benchmarks.** Several evaluation frameworks or benchmarks have been established for FL algorithms, each offering unique perspectives. For instance, LEAF [30] and TFF [31] tailor to heterogeneous datasets, whereas FedML [32], Flower [33], and FedScale [34] emphasize diverse system resources. Diverging from traditional tasks such as image classification and next word prediction, FedNLP [35] explores FL with challenging natural language processing applications, while FS-G [36] directs its attention to graph learning. pFL-Bench [37] offers a comprehensive evaluation of personalization in FL, while FL-bench [38] further focuses on domain generalization. Moreover, FLAIR [39] offers a curated large-scale dataset with fine-grained annotated labels and evaluated common FL baselines using it. Profit [40] is related to FLIP as it explores PL for personalization FL. However, they considered the PL of large language models rather than vision-language foundational models as considered in this paper. To summarize, the majority of the above mostly considers general FL algorithms, which cannot offer superior generalization performance as PL algorithms under data scarcity and distribution shifts. While it is crucial to test known hypotheses agreed upon by the community with standardized evaluations, FLIP presents a new and unique benchmark in anticipation of future research questions and challenges pertaining to FPL.

## 3 Problem Formulation

**Federated Learning.** We follow the widely-used federated averaging (FedAvg) [4] baseline as an example to introduce the FL problem setup. Specifically, it optimizes the global objective function $F(\boldsymbol{\theta})$ on local private data $\mathbb{D}_i$ with a total of $|\mathbb{C}|$ participating clients:

$$\min_{\boldsymbol{\theta} \in \mathbb{R}^d} F(\boldsymbol{\theta}) = \sum_{i=1}^{|\mathbb{C}|} \rho_i F_i(\boldsymbol{\theta}), \tag{1}$$

where $F_i(\boldsymbol{\theta}) = \frac{1}{|\mathbb{D}_i|} \sum_{\xi \in \mathbb{D}_i} f_i(\boldsymbol{\theta})$ is the local learning objective of client $i$, and $\sum_{i=1}^{|\mathbb{C}|} \rho_i = 1$. In each communication round, the server first transmits the updated global models to a selected group of clients. The selected clients perform the local training on private data and then transmit the trained model to the global server. FedAvg [4] aggregates the models by a weighted-averaging scheme based on the normalized number of data on each client. Without losing generality, we adapt several existing PL methods [11, 12, 18, 13, 17, 19, 15] to federated training with FedAvg [4].

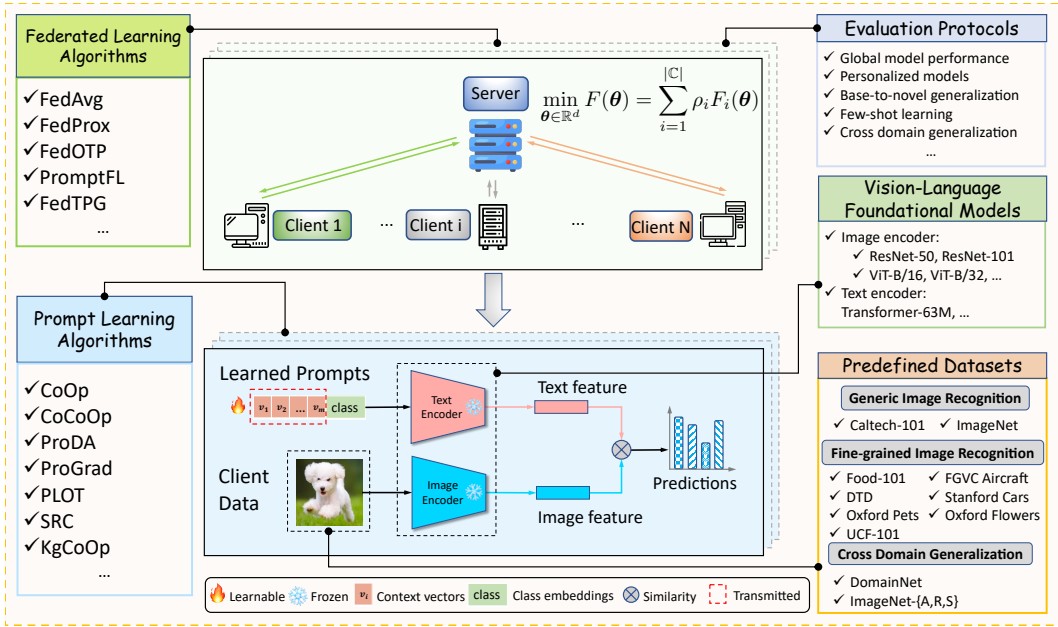

Figure 1: Overview of the FLIP framework.

**Federated Prompt Learning.** Following the success of vision-language pretrained models, recently the FL community has began to switch from train-from-scratch paradigm to adapting these models to diverse downstream tasks under federation. Prompt learning, a prevalent technique in NLP [41, 42, 43] has been investigated under vision applications [44]. Taking CLIP [9] for example, PL aims to adapt its frozen pretrained models, consisting of an image encoder $E_{\text{image}}$ and a text encoder $E_{\text{text}}$ for downstream tasks. In this benchmark, we seek to evaluate *continuous* PL, which optimizes a set of soft prompt vectors $\{v_1, v_2, \ldots v_m\}$ parameterized by a few learnable weights $\theta$. To obtain the text features of a class, we can concatenate the learned prompt vectors and the embedding features of a class name to construct the class-specific embeddings $t_j$, followed by feeding it to a text pretrained encoder $E_{text}$. For classification tasks of $n$ classes, the prediction probability of an input image $x$ for class $j$ can be calculated by comparing the similarity between encoded image features and text features of each class as follows:

$$p_{\boldsymbol{\theta}}(y = j \mid x) = \frac{\exp(\text{sim}(E_{\text{image}}(x), E_{\text{text}}(t_j))/\tau)}{\sum_i^n \exp(\text{sim}(E_{\text{image}}(x), E_{\text{text}}(t_i))/\tau)}, \qquad (2)$$

where $\text{sim}(\cdot, \cdot)$ represents a metric function such as dot product or cosine similarity, and $\tau$ is a scaling factor to control the temperature of Softmax operation. To optimize the prompt vectors, FPL enables the communication-efficient FL training by synchronizing the prompt parameters across clients under the coordination of a global server. Keeping true to conventional federated training which transmits trained parameters, FPL communicates the prompt-related parameters between the server and client for iterative optimization.

## 4 Benchmark Design

Figure 1 provides an overview of the FLIP framework design. This section details the models and 12 datasets used, 13 algorithmic baselines for FPL, and 3 evaluation protocols with various metrics under 6 evaluation scenarios.

### 4.1 Models and Datasets

**Models** All algorithms adopts pretrained CLIP [9] models as the base models for PL. CLIP models are vision-language models that consist of an image encoder $E_{\text{image}}$ and a text encoder $E_{\text{text}}$, both of which map the input data into a shared feature space. The architecture of the image encoder can either

be a ResNet-50 [45] or Vision Transformer (ViT) [46], whereas the text encoder adopts a transformer [47] architecture. In the main experiments, we used the CLIP model based on ResNet-50.

**Datasets** For the main evaluations, we used 8 publicly available datasets with annotated labels for global shared models $\alpha_g$ and personalized models $\alpha_p$: **Caltech**-101 [48], **DTD** [49], FGVC-**Aircraft** [50], **Food**-101 [51], Stanford**Cars** [52], Oxford-**Flowers**-102 [53], Oxford-**Pets** [54], and **UCF**-101 [55]. For novel class and few-shot scenarios, we use the first 4 datasets consisting of generic and fine-grained image recognition tasks. For brevity, we use the bolded part of the dataset names as the shorthand in the results. For cross-domain metrics, we further examine the robustness of FPL algorithms on distribution shifts, where the ImageNet [56] is used as the source domain, and the target domains include variants of ImageNet-derived datasets: **ImageNet-S**ketch [57] (images transformed to sketches) **ImageNet-A**dversarial [58] (natural adversarial examples in the wild), and **ImageNet-R**endition [59] (images with diverse styles).

## 4.2 Algorithmic Baselines

We include PromptFL [21], FedOTP [16], FedTPG [60], FedPGP [23], PromptFolio [22], and DP-FPL [29] as FPL baselines in our comparison. To further enrich baseline comparisons, we adapt a rich set of centralized prompt tuning algorithms mentioned in Section 2 to the federated setting to form each federated variant, prefixed with "$f$-", namely $f$-CoCoOp, $f$-PLOT, $f$-ProDA, $f$-ProGrad, $f$-SRC, and $f$-KgCoOp. While we provide other aggregation strategies in our codebase, for a fair comparison in this paper, all variants use federated averaging (FedAvg) [4] to aggregate prompt-related parameters on the server. Finally, we include ZS-CLIP, *i.e.*, the pretrained CLIP [9] as a zero-shot baseline (*i.e.*, with hand-crafted prompts) to provide a reference point.

## 4.3 Evaluation Protocols and Metrics

**Prompt learning protocols** We set the prompt context token length as 4. We try to align the number of prompts for methods evaluated. Specifically, we use a single set of prompt as default, with the exceptions that FedOTP, FedPGP, PromptFolio and DPFPL use two set of prompts for global and local representation. $f$-ProDA adopts two set of prompts for prompt distribution learning. For all evaluations, we fix the class token position in the *end* without token position augmentation. We only learn the prompt for the text input unless otherwise stated. For specific hyperparameter settings related to PL, please refer to Appendix B.4.

**FPL protocols** We evaluate the FPL algorithms under 3 protocols as follows. Each experiment was conducted with 3 runs with different random seeds. We report the best test accuracy on test set with mean value and standard deviation.

- **Standard learning** We simulate 10 clients with full participation to train a global model to evaluate on a shared test set. We use the standard SGD optimizer with initial learning rate 0.002, momentum 0.9 and a cosine learning rate decay scheduler to guarantee the convergence of each method. We set the batch size as 16, global communication rounds to 50, and the local training epoch to 1. For other detailed experimental hyperparameter settings under this protocol, please refer to Appendix B.4.

- **Partial participation** For the client sub-sampling protocol, we follow the standard learning above and increase the number of clients to 100 with a $10\%$ participation ratio to investigate the scalability of FPL algorithms.

- **Personalized learning** This protocol seeks to evaluate the adaptability of FPL methods to local data distribution for personalization purpose. Following [61], we split a test dataset sharing similar distribution of the training dataset for evaluation. The personalized learning protocol uses similar settings as standard learning, except that the performance is evaluated on personalized test sets instead of a shared test set.

**Evaluation scenarios** We use the following 6 scenarios where each reports evaluation metrics to assess the performance of FPL algorithms with *different data splits and evaluation settings*. Here, each scenario adopts the standard learning protocol unless otherwise stated.

- **Global shared FPL** (reports: $\alpha_g$) We split the training, validation and testing samples following [12]. We apply a Dirichlet data partition scheme with the concentration parameter

be set as 0.1 that produces non-*i.i.d.* data sets for local clients following [62]. This metric is evaluated under the standard learning and partial participation protocols.

- **Personalized FPL** (reports: $\alpha_{\rm p}$) The accuracy is reported by a weighted averaging of local clients' accuracy based on the number of data each client possesses. This is more robust than a direct averaging of local clients' accuracy as it reduces the impact of the potential fluctuating accuracy of some clients that hold very few training and testing samples. This metric is evaluated under the personalized learning protocol. For this scenario, we additionally evaluate methods that are designed for personalized FL, including FedPGP, PromptFolio, and DP-FPL.

- **Base-to-novel class generalization** (reports: $\alpha_{\rm b}$, $\alpha_{\rm n}$) As PL algorithms has the ability to generalize to unseen classes, we evaluate the FL variants on novel class generalization following the protocol below. First, we split all classes into two equal sets, where only the first half (containing base classes) is used for training, following the standard learning protocol with non-*i.i.d.* data heterogeneity. After training, we evaluate the model on the test sets of both halves (respectively containing base and novel classes). We report the accuracy on both sets, denoted as $\alpha_{\rm b}$ and $\alpha_{\rm n}$ respectively, along with the harmonic mean of the two accuracies, namely, $\alpha_{\rm h} \triangleq 2/(\alpha_{\rm b}^{-1} + \alpha_{\rm n}^{-1})$.

- **Few-shot learning** (reports: $\alpha_{\rm s=K}$) For few-shot generalization evaluation, where each client has only a small training shot $K$ per class, we apply *i.i.d.* sampling to draw $K$ samples per class for each client from the training set.

- **Cross-domain generalization** (reports: $\alpha_{\rm x \to y}$, where "x" and "y" denote the source and target domain datasets, respectively.) To evaluate the robustness of FPL algorithms on distribution shifts, we benchmark the performance on the cross-domain datasets with shared classes but distinctive domain distributions. Our training setting aligns with the standard learning protocol with non-*i.i.d.* data heterogeneity, using a source dataset (*e.g.*, ImageNet) for training, and a target dataset (*e.g.*, ImageNet-{A,R,S}) for testing.

- **Cost-performance trade-offs** (reports: global accuracy $\alpha_{\rm g}$, communication cost $\kappa$) Finally, certain PL-specific hyperparameters (*e.g.*, the number of prompts and prompt length) can influence the trade-off relationship between model performance and system resources, it is thus important to investigate how these hyperparameters affect the performance of FPL algorithms, in terms of both converged model performance, computational and communication costs. We conducted sensitivity analyses of various algorithms on these hyperparameters to explore both how they influence the trade-off relationship, and provide insights for sweet-spot configurations. We report the relevant results in Appendix C.2.

Beyond these accuracy metrics, we are also concerned with the stability of evaluated algorithms. Motivated by the "Ranking Scores" from the Out-of-Distribution Generalization literature [63, 64], we introduce a **superiority indicator**, which counts the total number of datasets on which a method's performance surpasses the baseline (PromptFL). The corresponding results are shown by columns with "#" header in each table. **Note that not all FPL methods are tailored to adapt to all scenarios listed above, therefore we evaluate each FPL method in their focused scenarios.** The details are outlined in Appendix B.3.

## 5   Experimental Results

We conduct extensive experiments under various FL scenarios. Below we present some important results on global and personalized performance, base-to-novel class generalization and few-shot learning capabilities. These metrics are primarily used in FL and PL for performance evaluation.

**Global shared FPL** In Table 1, we illustrate a comprehensive benchmarking of the performance of global models across 8 datasets. We summarize some key insights below:

1. After aligning the experimental settings, the performance gaps between the baseline PromptFL and various PL methods are less significant compared with the results reported in existing FPL literatures [21, 16]. Indeed, PromptFL [21], a simple combination of CoOp and FedAvg, serves as a very strong baseline for other FPL methods, occasionally achieving the best performance on fine-grained image recognition datasets such as OxfordPets and

Table 1: Comparison of **global shared model accuracy** $\alpha_g$ (%) of FPL methods. We report the mean $\pm$ standard deviation over 3 runs. The best and second-best results for each dataset are highlighted in **bold** and underlined, respectively. The "**#**" column indicates on how many datasets the method achieves performance exceeding **PromptFL**.

| Global $\alpha_g$ | Caltech | DTD | Aircraft | Food | Cars | Flowers | Pets | UCF | Avg. | # |
|---|---|---|---|---|---|---|---|---|---|---|
| **ZS-CLIP** | 86.0 | 41.7 | 16.6 | 77.9 | 55.5 | 65.3 | 85.7 | 61.5 | 61.3 | - |
| **PromptFL** | $91.5_{\pm0.5}$ | $57.6_{\pm1.3}$ | $22.8_{\pm0.4}$ | $79.2_{\pm0.1}$ | $62.0_{\pm0.4}$ | $\textbf{84.0}_{\pm1.7}$ | $\textbf{89.4}_{\pm0.5}$ | $70.1_{\pm0.8}$ | 69.6 | - |
| **FedOTP** | $91.8_{\pm0.1}$ | $58.0_{\pm0.8}$ | $21.9_{\pm0.4}$ | $78.7_{\pm0.1}$ | $62.8_{\pm0.2}$ | $83.3_{\pm0.6}$ | $89.1_{\pm0.1}$ | $69.4_{\pm0.6}$ | 69.4 | 3 |
| **FedTPG** | $90.2_{\pm0.1}$ | $56.8_{\pm1.0}$ | $19.0_{\pm1.2}$ | $79.3_{\pm0.2}$ | $60.7_{\pm0.2}$ | $78.0_{\pm1.6}$ | $89.0_{\pm0.6}$ | $68.3_{\pm0.2}$ | 67.6 | 1 |
| *f*-**CoCoOp** | $91.7_{\pm0.3}$ | $54.7_{\pm1.0}$ | $17.9_{\pm4.5}$ | $79.3_{\pm0.1}$ | $60.7_{\pm0.5}$ | $76.4_{\pm0.7}$ | $89.1_{\pm0.1}$ | $68.0_{\pm0.9}$ | 67.2 | 2 |
| *f*-**PLOT** | $91.6_{\pm0.3}$ | $\textbf{58.3}_{\pm1.4}$ | $21.7_{\pm0.5}$ | $78.3_{\pm0.2}$ | $60.7_{\pm0.8}$ | $83.4_{\pm0.6}$ | $88.9_{\pm0.5}$ | $69.7_{\pm0.4}$ | 69.1 | 2 |
| *f*-**ProDA** | $91.6_{\pm0.3}$ | $57.2_{\pm1.1}$ | $\textbf{23.1}_{\pm0.7}$ | $79.1_{\pm0.2}$ | $62.3_{\pm0.5}$ | $\textbf{84.0}_{\pm0.8}$ | $89.3_{\pm0.4}$ | $70.2_{\pm1.1}$ | 69.6 | 5 |
| *f*-**ProGrad** | $90.7_{\pm0.2}$ | $57.1_{\pm1.0}$ | $21.7_{\pm0.3}$ | $\textbf{79.5}_{\pm0.1}$ | $60.5_{\pm0.7}$ | $83.4_{\pm0.4}$ | $89.1_{\pm0.2}$ | $70.3_{\pm0.3}$ | 69.1 | 2 |
| *f*-**PromptSRC** | $\textbf{92.0}_{\pm0.8}$ | $57.8_{\pm0.3}$ | $21.2_{\pm0.4}$ | $78.6_{\pm0.4}$ | $62.4_{\pm0.2}$ | $83.6_{\pm0.1}$ | $89.2_{\pm0.7}$ | $70.3_{\pm1.0}$ | 69.4 | 4 |
| *f*-**KgCoOp** | $91.8_{\pm0.2}$ | $58.2_{\pm0.8}$ | $23.0_{\pm0.1}$ | $79.4_{\pm0.2}$ | $61.7_{\pm0.7}$ | $83.9_{\pm0.5}$ | $\textbf{89.4}_{\pm0.2}$ | $\textbf{70.4}_{\pm0.7}$ | **69.7** | **5** |

Table 2: Comparison of **personal model accuracy** $\alpha_p$ (%) of FPL methods on various datasets.

| Personal $\alpha_p$ | Caltech | DTD | Aircraft | Food | Cars | Flowers | Pets | UCF | Avg. | # |
|---|---|---|---|---|---|---|---|---|---|---|
| **ZS-CLIP** | 86.0 | 41.7 | 16.6 | 77.9 | 55.5 | 65.3 | 85.7 | 61.5 | 61.3 | - |
| **PromptFL** | $91.5_{\pm0.4}$ | $69.5_{\pm4.1}$ | $33.8_{\pm0.1}$ | $\textbf{82.1}_{\pm0.4}$ | $67.7_{\pm0.6}$ | $\textbf{89.7}_{\pm0.2}$ | $\textbf{89.9}_{\pm0.6}$ | $77.5_{\pm1.5}$ | 75.2 | 0 |
| **FedOTP** | $\textbf{91.9}_{\pm0.4}$ | $\textbf{73.8}_{\pm1.4}$ | $\textbf{36.1}_{\pm0.5}$ | $82.0_{\pm0.7}$ | $\textbf{68.1}_{\pm1.7}$ | $89.6_{\pm0.3}$ | $89.5_{\pm1.2}$ | $\textbf{80.7}_{\pm1.0}$ | **76.5** | 5 |
| **FedTPG** | $89.9_{\pm0.4}$ | $66.8_{\pm0.6}$ | $31.6_{\pm0.3}$ | $81.9_{\pm0.3}$ | $66.2_{\pm0.2}$ | $86.0_{\pm0.4}$ | $88.9_{\pm0.8}$ | $78.0_{\pm0.7}$ | 73.7 | 1 |
| **FedPGP** | $91.8_{\pm0.4}$ | $68.0_{\pm0.5}$ | $35.2_{\pm0.4}$ | $81.0_{\pm0.2}$ | $66.7_{\pm0.6}$ | $84.1_{\pm1.6}$ | $87.4_{\pm0.3}$ | $77.8_{\pm0.5}$ | 74.0 | 3 |
| **PromptFolio** | $91.6_{\pm0.3}$ | $70.2_{\pm0.4}$ | $34.6_{\pm0.5}$ | $82.0_{\pm0.3}$ | $67.4_{\pm0.2}$ | $89.2_{\pm0.5}$ | $89.4_{\pm0.2}$ | $79.2_{\pm0.6}$ | 75.5 | 4 |
| **DP-FPL** | $90.2_{\pm0.4}$ | $65.2_{\pm0.5}$ | $28.5_{\pm0.5}$ | $80.1_{\pm0.6}$ | $64.5_{\pm1.4}$ | $78.6_{\pm0.6}$ | $82.4_{\pm0.8}$ | $72.2_{\pm0.7}$ | 70.2 | 0 |
| *f*-**CoCoOp** | $91.8_{\pm0.4}$ | $70.3_{\pm3.0}$ | $34.0_{\pm1.7}$ | $81.8_{\pm0.6}$ | $67.4_{\pm0.7}$ | $86.4_{\pm1.9}$ | $89.5_{\pm0.9}$ | $77.1_{\pm0.9}$ | 74.8 | 3 |
| *f*-**PLOT** | $91.7_{\pm0.4}$ | $71.3_{\pm3.1}$ | $34.0_{\pm0.8}$ | $81.4_{\pm1.2}$ | $67.9_{\pm1.1}$ | $89.3_{\pm0.5}$ | $88.6_{\pm0.3}$ | $79.6_{\pm1.8}$ | 75.5 | 5 |
| *f*-**ProDA** | $91.7_{\pm0.9}$ | $69.7_{\pm2.6}$ | $34.7_{\pm0.6}$ | $\textbf{82.1}_{\pm1.3}$ | $67.8_{\pm1.3}$ | $89.3_{\pm0.6}$ | $\textbf{89.9}_{\pm0.5}$ | $77.9_{\pm1.5}$ | 75.4 | **6** |
| *f*-**ProGrad** | $91.7_{\pm0.6}$ | $69.2_{\pm0.3}$ | $32.9_{\pm0.7}$ | $81.6_{\pm0.8}$ | $67.0_{\pm1.0}$ | $88.8_{\pm0.9}$ | $89.5_{\pm0.7}$ | $77.0_{\pm1.5}$ | 74.7 | 1 |
| *f*-**PromptSRC** | $91.7_{\pm0.4}$ | $69.3_{\pm1.3}$ | $32.4_{\pm2.3}$ | $81.8_{\pm1.1}$ | $67.6_{\pm1.1}$ | $89.2_{\pm1.7}$ | $89.3_{\pm1.7}$ | $78.2_{\pm1.1}$ | 74.9 | 2 |
| *f*-**KgCoOp** | $91.6_{\pm0.3}$ | $68.6_{\pm2.7}$ | $31.3_{\pm0.5}$ | $81.4_{\pm0.7}$ | $67.1_{\pm1.4}$ | $88.9_{\pm0.9}$ | $\textbf{89.9}_{\pm0.3}$ | $76.9_{\pm0.9}$ | 74.5 | 1 |

Flowers over its competitors. We advocate acknowledging its simplicity and merits and including it as a reference baseline in all FPL works.

2. In most cases, the *f*-CoCoOp produces inferior results compared with PromptFL baseline. We hypotheses this is caused by its adoption of an image feature aggregation module, which is susceptible to the data heterogeneity raised by non-*i.i.d.* data partitions. As a result, its aggregated features may deviate from real class semantics if the biased local training is not properly counteracted. This underscores the potential risks of a direct porting of centralized PL methods to FL regime.

3. From the superiority indicator, we can observe the regularization-based FPL methods, such as *f*-SRC and *f*-KgCoOp generally produce discernible improvements for FL generic performance. It demonstrates such regularization can yield a favorable effect to reduce the local client drift [65]. Specifically, this regularization enforces all participating clients share a common objective that encourages the learning of domain knowledge by introduced a prescribed text prompt. This highlights the similar intuitions behind the regularization-based FL such as FedProto [66] and PL methods exemplified by PromptSRC [17].

**Personalized FPL** Table 2 presents the personalized performance comparison. Interestingly, we find FedOTP [16] generally outperforms other methods, emphasizing the potential of distribution alignment, for example, with Optimal Transport (OT) to adapt to personalized data distribution. The results also indicate FedOTP consistently outperforms a baseline method *f*-PLOT, which also applying OT to align representations across modalities, demonstrating advantage of *imbalanced* Optimal Transport over *f*-PLOT for personalized FL scenarios. This implies in addition to modality gap between vision and text representations, the distribution gap under FPL introduces additional challenges to be addressed. Besides, indicated by the superiority metric ("#"), we observe the improvements of regularization-based prompt learning methods under personalized data are less prominent compared with the results in Table 1. This could be an intrinsic dilemma that achieving

Table 3: **Base- and novel-class accuracy** (%) across 10 different splits of base and novel classes, "$\alpha_b$", "$\alpha_n$" and "$\alpha_h$" respectively denote base $\alpha_b$ and novel class $\alpha_n$ accuracies, and their harmonic mean $\alpha_h$.

| Metric | Caltech $\alpha_b$ | $\alpha_n$ | $\alpha_h$ | Aircraft $\alpha_b$ | $\alpha_n$ | $\alpha_h$ | Cars $\alpha_b$ | $\alpha_n$ | $\alpha_h$ | Flowers $\alpha_b$ | $\alpha_n$ | $\alpha_h$ | Avg. $\alpha_b$ | $\alpha_n$ | $\alpha_h$ | # |
|---|---|---|---|---|---|---|---|---|---|---|---|---|---|---|---|---|
| **ZS-CLIP** | 88.2 | 92.6 | 90.3 | 19.6 | 24.7 | 21.8 | 59.5 | 68.1 | 63.5 | 77.2 | 71.0 | 73.9 | 61.1 | 64.1 | 62.4 | - |
| **PromptFL** | 92.8±0.8 | 92.7±0.7 | 92.6±0.2 | 20.9±0.4 | 24.7±0.5 | 22.6±0.2 | 63.0±0.7 | 67.6±0.7 | 65.2±0.1 | 79.9±2.9 | 69.3±0.7 | 74.2±1.0 | 64.1 | 63.8 | 63.8 | - |
| **FedOTP** | 93.1±0.2 | 93.7±0.4 | 93.4±0.1 | 21.0±0.8 | 23.7±1.0 | 22.2±0.8 | 62.1±0.1 | 66.0±0.1 | 64.0±0.4 | 81.0±0.6 | 68.7±1.9 | 74.3±1.2 | 64.3 | 63.0 | 63.5 | 2 |
| **FedTPG** | 93.6±0.4 | 90.0±0.6 | 91.8±0.4 | 19.5±0.3 | 22.7±0.5 | 21.0±0.5 | 66.4±0.2 | 67.7±0.2 | 67.0±0.2 | 75.0±0.3 | 66.0±0.5 | 70.2±0.4 | 63.6 | 61.6 | 62.5 | 2 |
| **FedPGP** | 93.2±0.4 | 92.7±1.1 | 92.9±0.5 | 19.8±1.0 | 19.2±2.1 | 19.5±0.9 | 63.7±1.4 | 67.8±0.7 | 65.7±0.9 | 80.7±1.4 | 66.8±1.7 | 73.1±1.2 | 64.3 | 61.6 | 62.8 | 2 |
| ***f*-CoCoOp** | 92.7±0.8 | 93.6±0.6 | 93.1±0.2 | 18.0±2.1 | 17.1±2.4 | 17.2±2.2 | 62.8±0.6 | 66.7±0.3 | 64.7±0.2 | 79.4±1.4 | 70.7±1.8 | 74.8±0.6 | 63.2 | 62.0 | 62.4 | 2 |
| ***f*-PLOT** | 93.4±0.6 | 93.5±1.0 | 93.5±0.8 | 19.0±0.8 | 23.4±0.4 | 21.0±0.4 | 62.4±0.5 | 65.4±1.4 | 63.8±0.7 | 78.7±3.0 | 68.3±1.3 | 73.1±0.6 | 63.4 | 62.6 | 62.8 | 1 |
| ***f*-ProDA** | 93.0±0.4 | 92.7±1.1 | 92.8±0.4 | 22.0±0.5 | 25.1±1.1 | 23.4±0.6 | 63.3±0.7 | 67.7±0.4 | 65.4±0.2 | 77.9±0.7 | 69.6±0.5 | 73.5±0.1 | 64.0 | 63.8 | 63.8 | 3 |
| ***f*-ProGrad** | 93.2±0.4 | 93.0±0.4 | 93.1±0.4 | 21.6±0.5 | 25.2±1.6 | 23.3±0.4 | 64.2±0.6 | 67.9±0.5 | 66.0±0.3 | 80.3±1.1 | 70.6±0.5 | 75.2±0.2 | 64.7 | 64.2 | 64.3 | 4 |
| ***f*-SRC** | 90.2±0.2 | 93.0±0.1 | 91.6±0.1 | 21.7±1.0 | 24.9±1.0 | 23.1±0.8 | 61.2±0.2 | 67.1±0.3 | 64.0±0.1 | 78.9±1.1 | 70.2±0.6 | 74.3±0.7 | 62.3 | 63.8 | 62.8 | 3 |
| ***f*-KgCoOp** | 93.5±0.4 | 93.4±1.0 | 93.4±0.5 | 22.3±0.7 | 25.1±0.5 | 23.6±0.5 | 63.7±0.3 | 67.5±0.3 | 65.6±0.1 | 79.6±2.7 | 68.9±1.2 | 73.9±0.5 | 64.8 | 63.7 | 64.1 | 3 |

Table 4: Comparison of **few-shot accuracies** $\alpha_{s=k}$ (%) of FPL methods on Tiny-ImageNet. ZS-CLIP accuracy is 34.1%.

| $k =$ | 1 | 2 | 4 | 8 | 16 | Avg. | # |
|---|---|---|---|---|---|---|---|
| **PromptFL** | 39.9±0.4 | 42.1±0.2 | 44.0±0.2 | 45.8±0.8 | 47.0±0.2 | 43.7 | - |
| **FedOTP** | 41.0±0.2 | 43.9±0.1 | 46.4±0.2 | 48.2±0.2 | 49.4±0.1 | 45.8 | 5 |
| **FedTPG** | 40.1±0.6 | 41.9±0.4 | 42.4±0.2 | 44.2±0.4 | 46.4±0.2 | 41.8 | 1 |
| ***f*-CoCoOp** | 38.2±0.4 | 41.9±0.2 | 43.3±0.3 | 45.4±0.3 | 45.3±0.2 | 42.8 | 0 |
| ***f*-PLOT** | 39.5±0.2 | 41.7±0.1 | 43.2±0.4 | 45.3±0.4 | 46.3±0.1 | 43.2 | 0 |
| ***f*-ProDA** | 40.3±0.1 | 42.0±0.3 | 44.0±0.2 | 45.6±0.3 | 47.2±0.3 | 43.8 | 2 |
| ***f*-ProGrad** | 35.6±0.8 | 36.7±0.4 | 38.6±0.2 | 38.8±0.4 | 39.8±0.4 | 37.9 | 0 |
| ***f*-SRC** | 41.6±0.4 | 42.6±0.5 | 44.4±0.5 | 45.0±0.2 | 46.1±0.2 | 43.9 | 3 |
| ***f*-KgCoOp** | 39.6±0.4 | 42.0±0.6 | 43.6±0.9 | 46.3±0.2 | 46.5±0.3 | 43.6 | 1 |

Figure 2: **Few-shot accuracies on Tiny-ImageNet.**

improvements on global and personalized performance could compromise each other and requires further exploration.

**Base-to-novel generalization** We illustrated the results on base $\alpha_b$ and novel $\alpha_n$ class accuracies, along with their harmonic mean $\alpha_h$ in Table 3. Similar to global learning, Table 3 indicates regularization with generic prompt knowledge [17, 19, 15] prevents the client model from over-fitting the local data distribution, and it contributes to achieving the best trade-offs between base class fitting and novel class generalization. Different from the global benchmarking scenario, the naïve PromptFL is less effective in handling this challenging scenario according to the "**#**" competition metric.

**Few-shot generalization** Focusing on varying number of shots, Table 4 shows the few-shot training accuracies of FPL methods on Tiny-ImageNet. Notably, we sweep the number of shots with $k \in \{1, 2, 4, 8, 16\}$. Figure 2 also visualizes the few-shot training accuracies. Intriguingly, the results in Table 6 suggest that all three methods that achieve superior performance either use multiple prompt sets (FedOTP [16] and $f$-ProDA [13]) or sample multiple prompts ($f$-SRC [17]). These results hint the effectiveness of knowledge ensemble from multiple prompts, which reduce the *sample selection*

Table 5: Accuracy (%) under **client subsampling** with 10% of the total 100 clients.

| | Caltech | Aircraft | Cars | Flowers | Avg. | # |
|---|---|---|---|---|---|---|
| **ZS-CLIP** | 86.0 | 16.6 | 55.5 | 65.3 | 55.9 | - |
| **PromptFL** | 90.3±0.2 | 20.9±0.7 | 60.9±0.5 | 75.9±1.3 | 62.0 | - |
| **FedOTP** | 91.1±0.6 | 21.2±0.8 | 59.1±0.1 | 76.2±0.9 | 61.9 | 3 |
| **FedTPG** | 90.8±0.4 | 19.2±0.9 | 59.8±0.5 | 76.0±0.6 | 61.5 | 2 |
| ***f*-CoCoOp** | 90.4±0.5 | 17.5±1.6 | 59.7±0.5 | 73.9±0.7 | 60.4 | 1 |
| ***f*-PLOT** | 90.6±0.2 | 20.5±0.6 | 59.1±0.7 | 74.8±1.5 | 61.3 | 1 |
| ***f*-ProDA** | 90.8±0.4 | 21.7±0.4 | 61.0±0.6 | 75.0±0.7 | 62.1 | 3 |
| ***f*-ProGrad** | 90.7±0.1 | 22.2±0.6 | 60.3±0.5 | 74.6±0.2 | 61.9 | 2 |
| ***f*-SRC** | 90.6±2.0 | 21.9±1.4 | 61.5±0.2 | 76.2±0.8 | 62.6 | 4 |
| ***f*-KgCoOp** | 91.2±0.1 | 22.0±0.6 | 60.4±0.3 | 76.7±1.1 | 62.6 | 3 |

Table 6: Comparison of **few-shot training accuracy** $\alpha_{s=1}$ (%) on various datasets.

| | Caltech | Aircraft | Cars | Flowers | Avg. | # |
|---|---|---|---|---|---|---|
| **ZS-CLIP** | 86.0 | 16.6 | 55.5 | 65.3 | 55.9 | - |
| **PromptFL** | 88.7±0.3 | 17.3±1.0 | 55.7±0.2 | 65.3±1.2 | 57.2 | - |
| **FedOTP** | 89.8±0.4 | 17.8±1.2 | 56.8±0.2 | 65.6±0.8 | 57.5 | 4 |
| **FedTPG** | 89.2±0.2 | 18.2±0.6 | 56.2±0.4 | 65.2±0.4 | 57.3 | 3 |
| ***f*-CoCoOp** | 87.6±0.6 | 17.6±0.5 | 55.4±0.2 | 64.6±1.3 | 56.3 | 1 |
| ***f*-PLOT** | 87.5±0.7 | 17.6±0.6 | 55.5±0.1 | 65.7±1.3 | 56.6 | 2 |
| ***f*-ProDA** | 89.0±0.2 | 17.3±0.6 | 56.1±0.7 | 65.7±1.3 | 57.0 | 3 |
| ***f*-ProGrad** | 89.4±0.6 | 18.4±0.2 | 56.2±0.3 | 63.5±0.6 | 56.9 | 3 |
| ***f*-SRC** | 89.2±0.3 | 18.9±0.3 | 56.4±0.5 | 65.4±0.1 | 57.5 | 4 |
| ***f*-KgCoOp** | 88.4±0.4 | 18.3±0.8 | 56.4±0.1 | 62.9±0.6 | 56.5 | 2 |

*bias* raised by limited training samples [67, 68]. In short, FedOTP excels in few-shot learning through its cooperative global-local prompt design and distribution alignment.

**Client sub-sampling** In cross-device FL system, the partial participation of massive clients could have a detrimental forgetting effect on the global model due to their temporarily joining and quitting FL training. In Table 5, we explored this effect by scaling up to 100 simulated clients with a 10% participating ratio. We observe that under this scenario, the regularization-based FPL methods usually gain advantages over the baselines. This could be attributed to the regularization loss which introduces a common optimization objective among clients, alleviating the catastrophic forgetting effect.

**Feature shift heterogeneity** Table 7 reports performance under feature-shift heterogeneity among clients. Although collaborative training uniformly lifts the CLIP baseline, the gains under feature shift are remarkably slender than those on a regular dataset without such domain gaps (Table 1), underscoring the persisting difficulty of this scenario and the need for further investigation.

Table 7: Comparison of **domain-specific accuracy** (%) under feature shifts data heterogenity. The **Avg.** column denotes a weighted average of the accuracies of all domains based on their corresponding image counts.

| Feature Shift | Clipart | Infograph | Painting | Quickdraw | Real | Sketch | Avg. | # |
|---|---|---|---|---|---|---|---|---|
| **ZS-CLIP** | 54.8 | 40.9 | 48.8 | 6.0 | 77.7 | 49.3 | 44.6 | - |
| **PromptFL** | $59.6_{\pm0.2}$ | $45.6_{\pm0.3}$ | $\underline{53.7}_{\pm0.3}$ | $8.9_{\pm0.1}$ | $79.8_{\pm0.1}$ | $\mathbf{54.2}_{\pm0.2}$ | $\underline{48.1}_{\pm0.1}$ | - |
| **FedOTP** | $58.4_{\pm0.2}$ | $45.2_{\pm0.1}$ | $53.4_{\pm0.2}$ | $\underline{9.0}_{\pm0.1}$ | $79.2_{\pm0.1}$ | $53.2_{\pm0.1}$ | $47.7_{\pm0.1}$ | 1 |
| **FedTPG** | $59.8_{\pm0.1}$ | $45.8_{\pm0.3}$ | $53.6_{\pm0.2}$ | $8.5_{\pm0.2}$ | $79.9_{\pm0.3}$ | $54.0_{\pm0.3}$ | $47.9_{\pm0.3}$ | 3 |
| $f$-**CoCoOp** | $\mathbf{60.0}_{\pm0.1}$ | $\mathbf{46.1}_{\pm0.2}$ | $53.0_{\pm0.3}$ | $9.1_{\pm0.2}$ | $79.8_{\pm0.2}$ | $\mathbf{54.2}_{\pm0.2}$ | $\underline{48.1}_{\pm0.2}$ | 5 |
| $f$-**PLOT** | $58.5_{\pm0.3}$ | $44.8_{\pm0.2}$ | $53.0_{\pm0.1}$ | $9.0_{\pm0.4}$ | $79.2_{\pm0.1}$ | $53.3_{\pm0.1}$ | $47.6_{\pm0.1}$ | 1 |
| $f$-**ProDA** | $59.5_{\pm0.2}$ | $45.6_{\pm0.1}$ | $\mathbf{53.8}_{\pm0.2}$ | $9.0_{\pm0.2}$ | $79.6_{\pm0.1}$ | $54.0_{\pm0.3}$ | $48.0_{\pm0.1}$ | 3 |
| $f$-**ProGrad** | $58.8_{\pm0.2}$ | $44.5_{\pm0.2}$ | $52.5_{\pm0.1}$ | $7.5_{\pm0.2}$ | $\underline{80.0}_{\pm0.1}$ | $53.0_{\pm0.1}$ | $47.3_{\pm0.1}$ | 1 |
| $f$-**SRC** | $59.0_{\pm0.1}$ | $44.6_{\pm0.4}$ | $52.6_{\pm0.1}$ | $7.8_{\pm0.1}$ | $79.7_{\pm0.1}$ | $52.9_{\pm0.1}$ | $47.3_{\pm0.1}$ | 0 |
| $f$-**KgCoOp** | $\underline{59.9}_{\pm0.1}$ | $\underline{45.9}_{\pm0.2}$ | $53.6_{\pm0.1}$ | $8.8_{\pm0.1}$ | $\mathbf{80.2}_{\pm0.1}$ | $\underline{54.1}_{\pm0.1}$ | $\mathbf{48.2}_{\pm0.1}$ | $\underline{4}$ |

**Cross-domain generalization** Table 8 summarizes the cross-domain generalization performance when clients are trained on ImageNet and subsequently evaluated on test sets that exhibit pronounced domain shifts. On this challenging benchmark, $f$-CoCoOp and $f$-KgCoOp consistently attain the highest accuracy across most target domains. Their robustness can be attributed to two complementary design principles: (i) the injection of test-time image features into the prompt-generation process, and (ii) a self-regularization mechanism that anchors the learned prompts to semantically rich templates. These designs explicitly reduce the distributional disparity between training and testing environments.

Table 8: Comparing the **cross-domain performance** of FPL methods. Here, the source domain is ImageNet (IN), and the target domains are ImageNet-A(dversarial), -R(endition), and -S(ketch), respectively denoted as IN-A, IN-R, IN-S.

| Cross-domain $\alpha_{IN\rightarrow}$ | IN-A | IN-R | IN-S | Avg. | # |
|---|---|---|---|---|---|
| **ZS-CLIP** | 21.7 | 56.1 | 33.4 | 37.1 | - |
| **PromptFL** | $\mathbf{24.9}_{\pm0.4}$ | $58.2_{\pm0.3}$ | $35.6_{\pm0.6}$ | 39.6 | - |
| **FedOTP** | $23.8_{\pm0.3}$ | $58.3_{\pm0.8}$ | $35.2_{\pm0.4}$ | 39.1 | $\underline{1}$ |
| **FedTPG** | $24.5_{\pm0.4}$ | $58.6_{\pm0.6}$ | $35.7_{\pm0.3}$ | 39.6 | 2 |
| $f$-**CoCoOp** | $24.0_{\pm0.6}$ | $\mathbf{59.8}_{\pm1.1}$ | $\mathbf{36.0}_{\pm1.0}$ | $\mathbf{39.9}$ | 2 |
| $f$-**PLOT** | $23.8_{\pm0.7}$ | $57.5_{\pm0.3}$ | $34.6_{\pm0.6}$ | 38.6 | 0 |
| $f$-**ProDA** | $\underline{24.7}_{\pm1.7}$ | $58.3_{\pm0.6}$ | $35.6_{\pm0.8}$ | 39.5 | 1 |
| $f$-**ProGrad** | $23.5_{\pm1.1}$ | $58.3_{\pm0.5}$ | $35.5_{\pm1.2}$ | 39.1 | 1 |
| $f$-**SRC** | $24.0_{\pm0.7}$ | $\underline{58.7}_{\pm0.9}$ | $35.1_{\pm0.5}$ | 39.3 | 1 |
| $f$-**KgCoOp** | $\underline{24.7}_{\pm1.2}$ | $\underline{58.7}_{\pm1.4}$ | $\underline{35.9}_{\pm0.7}$ | $\underline{39.8}$ | 2 |

**More benchmark results** Due to space limitation, we present additional benchmark results of centralized training (Appendix C.1). Moreover, in Appendix C.2 we show that FLIP fully support the cost-performance trade-off exploration between accuracies and training costs, and provide benchmark results on the trade-off relationship. In Appendix C.3, we present the evaluation results with a Transformer (ViT-B/16) as the image encoder. To summarize, we present the key takeaways from our benchmark results:

> **Key Takeaways**
>
> - **Global shared FPL** PromptFL, combining CoOp and FedAvg, serves as a simple yet effective baseline and is competitive on fine-grained datasets like Oxford-Pets and Flowers.
> - **Personalized FPL** FedOTP generally outperforms other personalized methods, highlighting the efficacy of distribution alignment in adapting to personalized data.
> - **Base-to-novel Generalization** Regularization prevents overfitting and balances base and novel class metrics, without it (PromptFL) is less effective.
> - **Few-shot Generalization** Methods that use multiple prompts (*e.g.*, FedOTP, $f$-ProDA, $f$-SRC) perform best in few-shot scenarios, indicating ensembling helps reduce sample selection bias.
> - **Client Sub-sampling** The regularization-based methods alleviate catastrophic forgetting better.
> - **Feature Shift Heterogeneity** It remains challenging to counteract the adverse impact of feature shift for all evaluated FPL methods.
> - **Cross-domain Generalization** Test-time image feature injection and self-regularization contribute to improving the robustness against cross domain shift.
> - **Cost-performance Trade-offs** Extra learnable parameters (CoCoOp) do not necessarily improve performance but could be detrimental to computational and communication efficiency.

## 6   Conclusion

This paper presents FLIP, the first comprehensive evaluation for FPL algorithms for vision-language model adaptation. Through extensive experiments on various datasets and evaluation scenarios, we demonstrate the effectiveness of PL in federated settings, particularly in challenging scenarios characterized by data scarcity, unseen classes, and cross-domain distributional shifts. FLIP provides a standardized and extensible open-source codebase, complete with evaluation metrics and various open datasets, facilitating further research in this promising area. FLIP serves as a valuable tool for researchers and practitioners to explore the trade-offs between model performance and system resources in FPL, paving the way for the development of more efficient and effective algorithms.

**Limitations**   First, there is currently a lack of understanding of whether PL methods may impose higher or lower safety risks compared to conventional FL methods. Future research should thus investigate the safety implications of PL. Second, further research is required to examine the robustness of FPL methods against adversarial clients attempting to compromise the FL process through manipulated local data or corrupted model updates.

## Acknowledgments and Disclosure of Funding

This work is supported in part by Technology Development Fund of Macao S.A.R (FDCT) under 0074/2025/AMJ and 0123/2022/AFJ, National Natural Science Foundation of China (62376263), Guangdong Basic and Applied Basic Research Foundation (2023B1515130002), Natural Science Foundation of Guangdong (2024A1515030209), Shenzhen Science and Technology Innovation Commission (JCYJ20230807140507015). This work was carried out in part at SICC, which is supported by SKL-IOTSC, University of Macau.

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

# A Datasets

We evaluated FPL methods on 12 datasets, including generic image recognition, fine-grained image recognition and domain generalization tasks. These datasets are prevalent in the FL and PL literature, covering a wide range of scales, domains and partitions. For the detailed experimental setup steps of these datasets, please refer to the instructions from our project page. Below, we briefly summarize these datasets.

## A.1 Generic Image Recognition

For generic image recognition, we considered the following three datasets.

- The **Caltech-101** [48] contains 9,146 images divided into 101 distinct object categories, along with a background category. Each object category has between 40 to 800 images, with most categories having around 50 images. The images are of variable sizes and depict objects in various poses and viewpoints against different backgrounds.

- The **ImageNet** dataset [56] we used in our experiments contains over 1.2 million labeled images spread across 1,000 categories.

- The **Tiny-ImageNet** dataset is a downsized variant of the ImageNet dataset. It consists of 200 distinct classes, each containing 500 training images, 50 validation images, and 50 test images, summing up to 100,000 images in total.

## A.2 Fine-grained Image Recognition

For fine-grained image recognition, we evaluated 7 datasets:

- The **Describable Textures Dataset (DTD)** [49] is a collection of images specifically designed for studying texture recognition task. It contains 5,640 images categorized into 47 classes, each representing a distinct texture described by human-centric attributes like "bumpy," "striped," or "polka-dotted." Each class includes 120 images sourced from diverse environments, ensuring variability in appearance. The DTD is commonly used to develop and evaluate algorithms for recognizing and classifying textures based on their describable properties.

- The **FGVC Aircraft** dataset [50] is a specialized collection of images aimed at fine-grained visual classification of aircraft models. It includes 10,000 images of aircraft, encompassing 100 different aircraft model variants. Each image is annotated with detailed information such as aircraft type, variant, and manufacturer. This dataset is used to develop and evaluate algorithms that can distinguish between visually similar aircraft models, making it valuable for applications requiring precise classification within a narrowly defined category.

- The **Food-101** dataset [51] is a collection of images designed for food recognition tasks in computer vision. It contains 101 categories of food, with each category represented by 1,000 images, totaling 101,000 images. The images are split into a training set of 75,750 images and a test set of 25,250 images. This dataset is used to develop and test algorithms that can accurately identify different types of food, making it useful for applications in dietary tracking, culinary automation, and food-related research.

- The **Oxford Pets** dataset [54] is a collection of images for the fine-grained classification and segmentation of pet breeds. It includes 37 categories, encompassing different breeds of cats and dogs, with roughly 200 images per category. Each image is annotated with breed labels and additional information like pixel-level segmentation masks. This dataset is used to develop and evaluate algorithms for pet breed identification and object segmentation.

- The **Oxford Flowers** dataset [33] is a collection of images designed for flower classification tasks. It comprises 102 flower categories, with each category containing between 40 and 258 images, totaling 8,189 images. The images are annotated with class labels, making the dataset suitable for automated botanical identification.

- **Stanford Cars** [52] includes 16,185 images of 196 car models, covering a variety of makers, models, and years. The dataset is divided into 8,144 training images and 8,041 testing images.

- **UCF Action Recognition** dataset [55] is a widely used dataset for action recognition in videos. It consists of thousands of video clips collected from YouTube across 101 action categories, such as walking, running, and jumping. In our experiments, we use the static frames of each action for PL, and evaluate the model performance on a disjoint test set.

## A.3 Domain Generalization

To evaluate the domain generalization ability of these algorithms, we optimize the prompt on ImageNet and further evaluate the obtained model on three datasets with domain shifts:

- The **ImageNet-A(dvarsarial)** dataset [58] introduces 7,500 *testing* images for 200 ImageNet classes. It contains real-world, unmodified, and naturally occurring examples that can significantly degrade the performance of machine learning models.

- The **ImageNet-R(endition)** dataset [59] consists of the renditions of ImageNet images such as art, cartoons and graffiti. It contains 200 ImageNet classes, with each class 150 images, resulting in total 30,000 images.

- The **ImageNet-S(ketch)** dataset [57] comprises 50,889 images, with about 50 images corresponding to each of the 1,000 ImageNet classes. These images are obtained through Google Image searches using the query "sketch of a <class>". The search is restricted to the "black and white" color scheme. Initially, 100 images are queried for each class, followed by manual curation to remove irrelevant and similar images. In cases where fewer than 50 images remain after cleaning, the dataset is augmented through image flipping and rotation.

As ImageNet-A and ImageNet-R contain only a fractional of classes from the original ImageNet dataset, we curated a subset of ImageNet that contains the images belonging to the 200 classes of training data. For ImageNet-S dataset, we use all 1,000 classes of ImageNet as the training data. Examples of these datasets are shown in Figure 3.

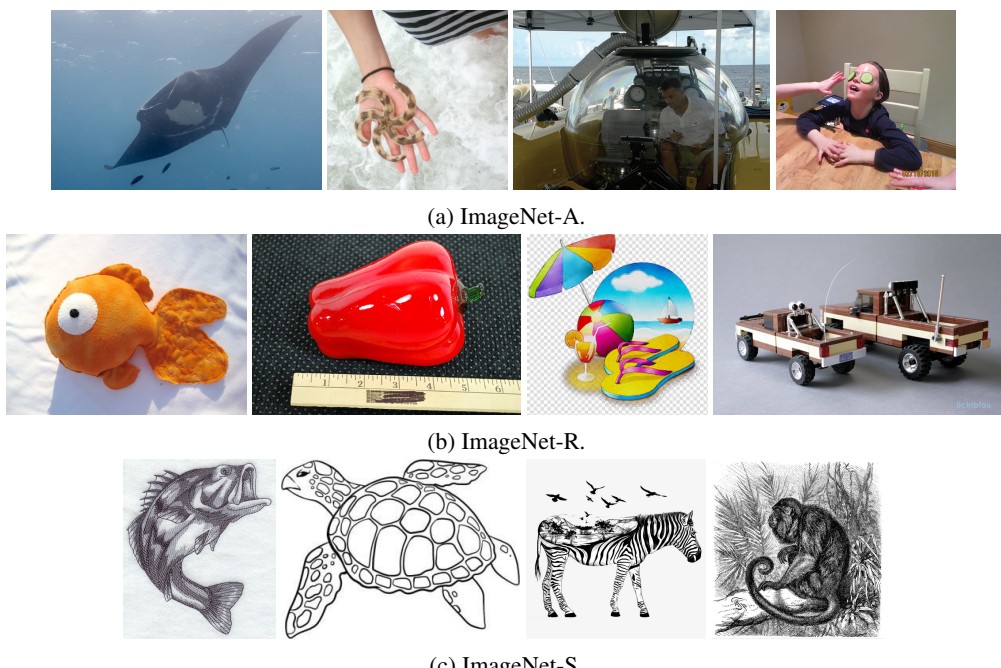

(a) ImageNet-A.

(b) ImageNet-R.

(c) ImageNet-S.

Figure 3: Example images sampled from ImageNet-A, ImageNet-R, and ImageNet-S.

# B Training Details

## B.1 Models

To align with previous works [21], we adapt the pretrained models from CLIP [9] to FPL. Specifically, we use a ResNet-50 model as the image feature encoder and a Transformer [69] model with 63 million parameters and 8 attention heads as the textual feature encoder. We freeze the weights of image and text encoders and only tune the learnable prompt for *text input*.

## B.2 Data Heterogeneity

### B.2.1 Feature-Shift Data Heterogeneity

In a FL system, the participating clients may collect data from distinct domains. This introduce the training-time feature shift data heterogeneity that could hinder the generalization of obtained models. In light of this, we evaluate the resilience of FPL algorithms under such data heterogeneity in addition to label distribution data heterogeneity. We use the **DomainNet** [70] dataset consisting of six domains, each representing a distinct visual domain such as **Clipart**, **Painting**, **Real**, **Quickdraw**, **Infograph**, and **Sketch**. Figure 4 exemplifies the images sampled from these domains. For each domain, we assign the training data to two clients, resulting in 12 clients with each client only possessing training data from a single domain.

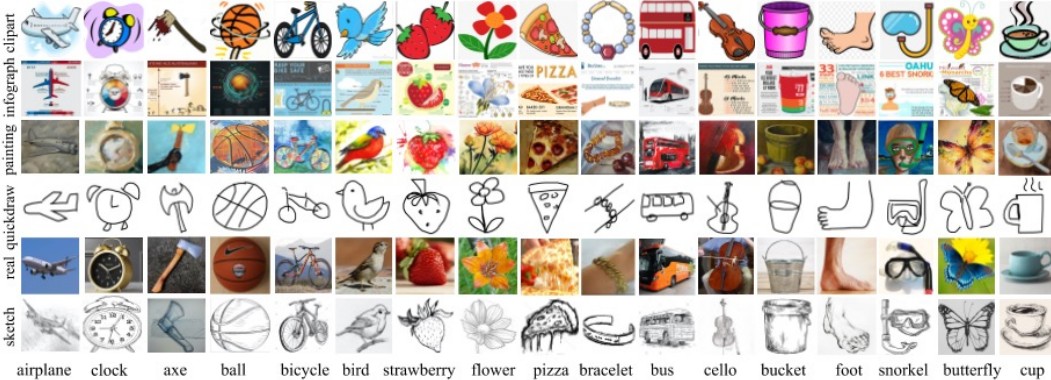

Figure 4: An illustration of images sampled from DomainNet [70][3].

### B.2.2 Class-Shift Data Heterogeneity

We adopt two **data partition strategies** for local clients, *i.e.*, the IID and non-IID, according to FL scenarios or metrics we evaluated:

- The **IID** dataset partition is only applied to evaluate the fewshot generalization performance of global models. Concretely, we assign each client equal training samples of each class based on the number of shots to evaluate the performance under limited training data.

- The **non-IID** dataset partition is applied to evaluate the performance of methods under label distribution data heterogeneity. We take the partition strategy from [62] to simulate heterogeneous data partition based on a Dirichlet distribution controlled by a concentration parameter $\alpha$. A smaller $\alpha$ indicates more aggressive data heterogeneity, while an infinitely large $\alpha$ degenerates to the *i.i.d.* data partition. We set $\alpha = 0.1$ by default in all experiments unless otherwise stated. To partition the original training data for multiple clients, we first subsample a smaller balanced dataset from the original training set, and then apply the Dirichlet data partition. For all experiments, we subsample 8 images for each class to construct the dataset for partition except for few-shot experiments mentioned above. Besides, we set 16 images for each class for partial participation scenarios.

---

[3] https://ai.bu.edu/M3SDA/.

Notably, the experimental setting on evaluating DomainNet introduces additional feature shift data heterogeneity raised by domain-specific features on each client. This can be considered as an extension on the non-*i.i.d.* data partition with domain heterogeneity.

## B.3 Methods

We consider the following methods as baselines of FPL. For more comprehensive comparison, we also include centralized methods, such as CoCoOp, PLOT, ProDA, ProGrad, Prompt-SRC and KgCoOp. All centralized methods are implemented for local client training and combined with FedAvg [4] for global aggregation. The evaluation helps to understand the properties of existing PL methods under a broad range of evaluation metrics for federated training. This also conveys insights of the appropriate application scenarios of each method under federation.

Below we make a brief introduction of FL methods:

- **PromptFL** [21] is a simple yet effective FPL method that can be viewed as a federated variant of CoOp [11] with FedAvg [4] for global aggregation. PromptFL only communicates the shared soft prompts instead of a shared global model as in conventional FL. This drastically reduces the communication cost of FL. We evaluate PromptFL in all experiments as an important baseline.

- **FedOTP**[4] [16] extends upon the PLOT [18] and designs a novel optimization scheme for imbalanced optimal transport. It also proposes to learn both local and global aligned representation for better generalization. The FedOTP is originally designed for personalized FL. To evaluate its performance on generic FL, we make a small alteration to allow all local prompt parameters to be communicated and updated. This allows the evaluation of FedOTP in most FPL scenarios.

- **FedTPG**[5] [60] designs a text-driven prompt generation network, which is conditioned on task-related text input, enabling robust generalization to both seen and unseen classes.. We evaluate FedTPG in most FPL scenarios except for cost-performance trade-off due to its employment of attention modules, which incur significantly larger communication overhead than other methods.

- **FedPGP**[6] [23] strikes a balance between personalization and generalization of FPL via low-rank adaptation and contrastive learning. As it focuses on personalized performance and generalization capability of client models. We evaluate it in personalized PFL and base-to-novel generalization scenarios.

- **PromptFolio**[7] [22] introduces a portfolio consisting of global prompt and local prompt to balance the generalization and personalization, motivated by portfolio optimization. This work also establishes a theoretical analysis framework for FPL based on feature learning theory. We evaluate it in the personalized FPL scenario.

- **DP-FPL**[8] [29] leverages global and local differential privacy to achieve a privacy-preserving personalized FPL approach for multi-modal LLMs. We evaluate it in the personalized FPL scenario.

In addition, we also evaluate a rich set of centralized prompt learning methods by seamlessly adapting them to FPL as a *local training methods* on clients. These methods are comprehensively evaluated in all FPL scenarios with the exception of MaPLe [71], which requires tuning the image and textual prompts in their corresponding Transformer models. Therefore, we only evaluate it on FPL scenarios with the vision Transformer (ViT-B/16) as the image encoder (Appendix C.3). Below are the details of evaluated algorithms:

- **CoCoOp**[9] [12] addresses the base- and novel-classes generalization dilemma of PL by introducing conditional inference. It optimizes an additional meta-net to deliver the im-

---

[4]`https://github.com/HongxiaLee/FedOTP`.

[5]`https://github.com/boschresearch/FedTPG`.

[6]`https://github.com/TianyuCui0v0/FedPGP`

[7]`https://github.com/PanBikang/PromptFolio`

[8]`https://github.com/linhhtran/DP-FPL`

[9]`https://github.com/KaiyangZhou/CoOp`.

age features for complementing the domain specific information during inference, which alleviates the over-fitting issue of the CoOp [11].

- **PLOT**[10] [18] introduces optimal transport to match the image and textual features. This benefits the distribution alignment of cross-modality features to reduce the modality gaps.

- **ProDA**[11] [13] proposes a optimization framework to learn a Gaussian distribution over possible prompts rather than relying on a single static prompt. It also prompts the diversity of prompt sets by introducing a orthogonality regularization loss term.

- **ProGrad**[12] [19] updates the prompt with aligned gradient (or non-conflicting) to the general knowledge which is achieved by regularizing gradient update with tailored prompts for domain-specific dataset.

- **KgCoOp**[13] [15] is a concurrent work that also introduces tailored prompts for each dataset as an anti-overfitting technique for guiding the prompt optimization. This reduces the discrepancy between the textual features produced by learnable prompts and the hand-crafted prompts, enhancing the generalization ability for unseen classes.

- **SRC**[14] [17] regularizes the PL with the predictions of the frozen model, multiple prompts over the training trajectory and textual diversity from different prompt templates. It reduces the catastrophic forgetting of generalizable knowledge from the pretrained CLIP models.

- **MaPLe**[15] [71] promotes better vision-language alignment on downstream tasks by introducing multi-modal PL. It also employs a coupling function to condition vision prompts on language counterparts, acting as a bridge between two modalities. We evaluate MaPLe on FPL scenarios with the vision Transformer (ViT-B/16) as the image encoder.

To unify the experimental settings for fair and faithful results, we adapt the official public code implementation of these centralized methods into our FL framework with minimal alterations such as renaming their original arguments. For methods that lacked public code, we either re-implement their algorithms or port the unofficial code with careful scrutinizing of the algorithmic details to ensure alignment with original papers. We will continuously include more FPL methods into FLIP.

## B.4 Hyperparameters

### B.4.1 General Hyperparameter Settings

We use the standard SGD optimizer with initial learning rate 0.002, momentum 0.9, and a cosine learning rate decay scheduler to guarantee the sufficient convergence of each method. We set the batch size as 16, global communication rounds 50 and the local training epoch 1. For each run, we use *random* prompt initialization without prompt position augmentation when constructing the entire prompt with prefix, class name and suffix. The input images is resized to $240 \times 240$ then cropped with size $224 \times 224$ to match the input image size of CLIP [9] image encoder, followed by random horizontal flipping and normalization. For each experiments, we conduct 3 independent runs with different random seeds and report the mean and standard variance of accuracy on the test set. Notably, Zhou *et al.*[12] splits the base and novel classes based on sorted class names (in alphabetical order). To rigorously benchmark FPL algorithms across different base and novel class splits, we expand the base-to-novel setup with nine random partitions plus the one from [12]. We report the averaged accuracy obtained from these different base and novel dataset partitions.

### B.4.2 Hyperparameters for FPL Algorithms

In addition to above general hyperparameters shared by all evaluated algorithms, we also conducted hyperparameter tuning for each algorithm in our benchmark. The goal was to identify optimal configurations that maximize performance while maintaining consistency across different experimental settings. While enforcing equal hyperparameter tuning budgets across all algorithms is critical for

---

[10]https://github.com/CHENGY12/PLOT.

[11]https://github.com/bbbdylan/proda (unofficial).

[12]https://github.com/BeierZhu/Prompt-align.

[13]https://github.com/htyao89/KgCoOp.

[14]https://github.com/muzairkhattak/PromptSRC.

[15]https://github.com/muzairkhattak/multimodal-prompt-learning.

fairness, disparities in the number and complexity of tunable parameters per algorithm introduce challenges in comparative evaluation. Specifically, some algorithms have no tunable hyperparameters other than the general hyperparameters shared by all algorithms, while others may occupy multiple extra hyperparameters. Moreover, the total computational cost grows exponentially *w.r.t.* the total number of hyperparameters of a FPL algorithm, making exhaustive hyperparameter tuning prohibitively expensive. To address this, we adopt a fixed tuning budget for each algorithm with a set of hyperparameter variants similar to those reported in their original papers. We report the averaged results of 3 experimental runs under the best hyperparameter configuration. In Table 9, we detail the hyperparameters explored for each algorithm.

Table 9: The hyperparameters explored for evaluated FPL algorithms.

| Algorithms | Hyperparameters | Specification |
|---|---|---|
| **FedOTP** | $\gamma \in [0.7, 0.8, 0.9]$ | hyperparameter for unbalanced OT |
| **FedPGP** | $\mu \in [0.5, 1, 5]$ | tradeoff parameter for the contrastive loss |
| **PromptFolio** | $\theta \in [0.1, 0.2, 0.4]$ | balancing coefficient |
| **DP-FPL** | $C_{th} \in [5, 10, 20]$ | clipping threshold |
| **$f$-PLOT** | $\lambda \in [0.01, 0.1, 1]$ | Entropy regularization hyperparameter for OT |
| **$f$-ProDA** | $\lambda \in [0.01, 0.1, 0.5]$ | tradeoff parameter for semantic orthogonality |
| **$f$-ProGrad** | $\lambda \in [0.4, 0.8, 1]$ | tradeoff parameter for gradient regularization |
| **$f$-SRC** | $\lambda_1, \lambda_2 \in [(1, 1), (5, 10), (10, 25)]$ | balancing coefficients for regularization losses |
| **$f$-KgCoOp** | $\lambda \in [1, 4, 8]$ | tradeoff parameter of the regularization loss |

## C    Additional Results

### C.1    Centralized Setting

In Table 10, we report the training accuracy values of PL methods under the centralized setting. *With the initialization of the pretrained models, there is only a slender margin between centralized and federated settings.* We speculate the underlying reason is that rich features from the pretrained models significantly reduce the potential gradient conflict among client updates. This observation holds the promise of closing the gap between centralized and federated training, motivating practical and efficient algorithms that specifically seek out better generalization with pretrained vision-language models.

Table 10: Comparison of training accuracy (%) of PL methods **under the centralized (*i.e.* non-FL) setting.** We report the mean $\pm$ standard deviation over 3 runs.

| Centralized | **Caltech** | **DTD** | **Aircraft** | **Food** | **Cars** | **Flowers** | **Pets** | **UCF** |
|---|---|---|---|---|---|---|---|---|
| **ZS-CLIP** | 86.0 | 41.7 | 16.6 | 77.9 | 55.5 | 65.3 | 85.7 | 61.5 |
| **CoOp** | $91.5_{\pm0.8}$ | $58.1_{\pm1.0}$ | $23.5_{\pm0.8}$ | $79.3_{\pm0.3}$ | $63.0_{\pm0.2}$ | $86.4_{\pm0.1}$ | $89.3_{\pm0.5}$ | $70.7_{\pm0.3}$ |
| **CoCoOp** | $91.9_{\pm0.2}$ | $57.2_{\pm1.0}$ | $19.1_{\pm1.4}$ | $79.4_{\pm0.5}$ | $62.7_{\pm0.2}$ | $79.9_{\pm1.5}$ | $88.9_{\pm0.2}$ | $68.7_{\pm1.5}$ |
| **PLOT** | $91.7_{\pm0.3}$ | $58.8_{\pm0.4}$ | $23.4_{\pm0.8}$ | $78.3_{\pm0.1}$ | $62.4_{\pm0.6}$ | $86.1_{\pm0.2}$ | $89.6_{\pm0.3}$ | $71.0_{\pm0.1}$ |
| **ProDA** | $91.8_{\pm0.3}$ | $57.0_{\pm0.8}$ | $22.8_{\pm0.2}$ | $79.0_{\pm0.2}$ | $63.6_{\pm0.6}$ | $88.6_{\pm0.7}$ | $89.0_{\pm0.2}$ | $70.9_{\pm0.5}$ |
| **ProGrad** | $91.2_{\pm0.3}$ | $57.8_{\pm1.0}$ | $21.7_{\pm1.3}$ | $79.4_{\pm0.1}$ | $63.3_{\pm0.1}$ | $87.9_{\pm0.3}$ | $89.1_{\pm1.2}$ | $70.1_{\pm0.9}$ |
| **PromptSRC** | $92.2_{\pm0.1}$ | $57.9_{\pm1.6}$ | $22.7_{\pm0.2}$ | $78.9_{\pm0.1}$ | $63.5_{\pm0.2}$ | $84.2_{\pm3.2}$ | $89.4_{\pm0.1}$ | $71.5_{\pm0.3}$ |
| **KgCoOp** | $91.8_{\pm0.2}$ | $58.6_{\pm0.5}$ | $23.8_{\pm0.1}$ | $79.5_{\pm0.3}$ | $64.3_{\pm0.6}$ | $84.3_{\pm2.0}$ | $89.6_{\pm0.7}$ | $71.3_{\pm0.7}$ |

### C.2    Cost-performance Trade-offs

Tables 11 and 12 present the communication and performance trade-off by changing the number of prompts or prompt context token length.    Figures 5a and 5b present the communication and performance trade-offs by changing the number of prompts and prompt token lengths respectively. First, we note that a direct scaling of the learnable parameters does not necessarily deliver positive improvements. For example, $f$-CoCoOp employs a meta-net to aggregate the conditional image

Table 11: **Trade-offs between accuracy (%) and the number of communicated parameters (in millions) under different number of prompts on Caltech.** Here, we sweep the number of prompts with $\{1, 2, 4\}$ while keeping the number of prompt tokens fixed at 4.

| Number of Prompts | 1 Accuracy | Cost | 2 Accuracy | Cost | 4 Accuracy | Cost | Avg. | # |
|---|---|---|---|---|---|---|---|---|
| **PromptFL** | $91.5_{\pm 0.5}$ | 2.05 | $91.4_{\pm 0.4}$ | 4.10 | $91.7_{\pm 0.1}$ | 8.19 | 91.6 | - |
| **FedOTP** | $\underline{91.8}_{\pm 0.1}$ | 4.10 | $\underline{91.8}_{\pm 0.5}$ | 8.19 | $\underline{91.9}_{\pm 0.3}$ | 16.38 | $\underline{91.8}$ | **3** |
| $f$-**CoCoOp** | $91.7_{\pm 0.3}$ | 100.93 | $91.3_{\pm 0.3}$ | 102.98 | $91.7_{\pm 0.1}$ | 107.07 | 91.5 | 1 |
| $f$-**PLOT** | $91.6_{\pm 0.3}$ | 2.05 | $91.2_{\pm 0.3}$ | 4.10 | $91.4_{\pm 0.2}$ | 8.19 | 91.4 | 1 |
| $f$-**ProDA** | $91.6_{\pm 0.3}$ | 4.10 | $91.1_{\pm 0.1}$ | 8.19 | $91.7_{\pm 0.1}$ | 16.38 | 91.5 | 1 |
| $f$-**ProGrad** | $90.7_{\pm 0.2}$ | 2.05 | $91.1_{\pm 0.1}$ | 4.10 | $91.4_{\pm 0.1}$ | 8.19 | 91.1 | 0 |
| $f$-**SRC** | $\mathbf{92.0}_{\pm 0.8}$ | 2.05 | $\mathbf{92.0}_{\pm 0.3}$ | 4.10 | $\mathbf{92.1}_{\pm 0.1}$ | 8.19 | $\mathbf{92.0}$ | 3 |
| $f$-**KgCoOp** | $\underline{91.8}_{\pm 0.2}$ | 2.05 | $91.4_{\pm 0.2}$ | 4.10 | $\underline{91.9}_{\pm 0.2}$ | 8.19 | 91.7 | $\underline{2}$ |

Table 12: **Trade-offs between accuracy (%) and the number of communicated parameters (in millions) under different number of prompt tokens on Caltech.** Here, we sweep the **number of tokens** with $\{4, 8, 16\}$ while keeping the number of prompts fixed at 1.

| Number of Tokens | 4 Accuracy | Cost | 8 Accuracy | Cost | 16 Accuracy | Cost | Avg. | # |
|---|---|---|---|---|---|---|---|---|
| **PromptFL** | $91.5_{\pm 0.5}$ | 2.05 | $91.0_{\pm 0.6}$ | 4.10 | $91.7_{\pm 0.2}$ | 8.19 | 91.4 | - |
| **FedOTP** | $\underline{91.8}_{\pm 0.1}$ | 4.10 | $\underline{91.8}_{\pm 0.2}$ | 8.19 | $\underline{92.0}_{\pm 0.3}$ | 16.38 | $\underline{91.8}$ | **3** |
| $f$-**CoCoOp** | $91.7_{\pm 0.3}$ | 100.93 | $91.6_{\pm 0.9}$ | 102.98 | $91.8_{\pm 0.2}$ | 107.07 | 91.7 | **3** |
| $f$-**PLOT** | $91.6_{\pm 0.3}$ | 2.05 | $91.4_{\pm 0.3}$ | 4.10 | $91.7_{\pm 0.1}$ | 8.19 | 91.6 | $\underline{2}$ |
| $f$-**ProDA** | $91.6_{\pm 0.3}$ | 4.10 | $91.5_{\pm 0.6}$ | 8.19 | $91.8_{\pm 0.2}$ | 16.38 | 91.6 | **3** |
| $f$-**ProGrad** | $90.7_{\pm 0.2}$ | 2.05 | $91.0_{\pm 0.3}$ | 4.10 | $91.6_{\pm 0.1}$ | 8.19 | 91.1 | 1 |
| $f$-**SRC** | $\mathbf{92.0}_{\pm 0.8}$ | 2.05 | $\mathbf{92.1}_{\pm 0.3}$ | 4.10 | $\mathbf{92.2}_{\pm 0.1}$ | 8.19 | $\mathbf{92.1}$ | **3** |
| $f$-**KgCoOp** | $\underline{91.8}_{\pm 0.2}$ | 2.05 | $91.2_{\pm 0.4}$ | 4.10 | $91.4_{\pm 0.1}$ | 8.19 | 91.5 | $\underline{2}$ |

information, which drastically increases the number of communications. However, this does not translates to accuracy boost over the simple baseline in most experiments. Besides, by comparing the accuracies increments of a single methods with different number of prompt or prompt length, we can conclude that methods with distribution alignment ($f$-FedOTP) or diversity regularization ($f$-ProDA) usually bring in stable improvements when scaling up the prompt parameters. Finally, both approaches for tweaking the prompt parameters yield similar improvements. Indeed, we do not observe clear dominance of them over the other.

### C.3 Evaluation on Transformer Image Encoder

In Tables 13 to 15, we respectively report the global, personal and base-to-novel accuracy metrics for the ViT-B/16 image encoder, following the evaluation protocols used in Tables 1 to 3. To sum up, these results show a similar trend on those of ResNet-50, and the ViT-B/16 in most experiments delivers better results than the ResNet-50 image encoder. Notably, $f$-MaPLe manifests clear advantages because of its multi-modal prompt optimization in both image and textual encoders, serving as a strong competitor compared with other federated prompt learning algorithms.

## D Discussion

### D.1 Implementation Details

**Environments** We implement all evaluated methods with PyTorch [72] of version 2.1.0. We try to minimize the number of packages used in our code framework, and setting up the environment only requires minutes. To alleviate computational burden, we apply the automatic mixed precision (AMP)

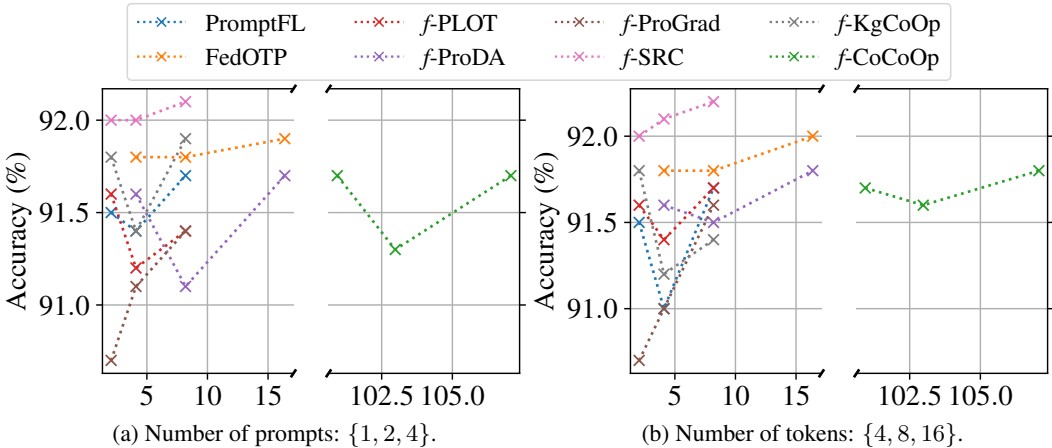

(a) Number of prompts: $\{1, 2, 4\}$.          (b) Number of tokens: $\{4, 8, 16\}$.

Figure 5: **Trade-offs between accuracy (%) and the number of communicated parameters (in millions) on Caltech-101.**

Table 13: **Comparison of shared global model accuracy $\alpha_g$ (%) of FPL methods with a ViT-B/16 image encoder.** Results are reported in a similar style as Table 1.

| Global $\alpha_g$ | Caltech | DTD | Aircraft | Food | Cars | Flowers | Pets | UCF | Avg. | # |
|---|---|---|---|---|---|---|---|---|---|---|
| **ZS-CLIP** | 93.5 | 45.0 | 24.3 | 85.5 | 65.6 | 68.0 | 89.2 | 67.5 | 67.3 | - |
| **PromptFL** | $95.5_{\pm0.1}$ | $59.6_{\pm0.7}$ | $31.2_{\pm0.3}$ | $86.8_{\pm0.1}$ | $70.2_{\pm1.3}$ | $\underline{86.2}_{\pm1.7}$ | $92.4_{\pm0.2}$ | $\underline{77.5}_{\pm0.6}$ | 74.9 | - |
| **FedOTP** | $\underline{95.6}_{\pm0.1}$ | $\underline{60.6}_{\pm0.7}$ | $\mathbf{32.4}_{\pm0.6}$ | $86.9_{\pm0.2}$ | $\underline{70.6}_{\pm0.5}$ | $85.6_{\pm0.3}$ | $92.2_{\pm0.5}$ | $76.6_{\pm0.6}$ | $\underline{75.1}$ | $\underline{5}$ |
| **FedTPG** | $95.4_{\pm0.2}$ | $60.2_{\pm0.3}$ | $31.4_{\pm0.2}$ | $86.8_{\pm0.2}$ | $70.2_{\pm0.1}$ | $85.7_{\pm0.3}$ | $91.8_{\pm0.1}$ | $76.2_{\pm0.3}$ | 74.8 | 3 |
| $f$-**CoCoOp** | $\underline{95.6}_{\pm0.2}$ | $57.7_{\pm1.0}$ | $31.0_{\pm0.6}$ | $86.6_{\pm0.2}$ | $68.5_{\pm0.3}$ | $81.6_{\pm0.8}$ | $92.4_{\pm0.7}$ | $74.5_{\pm0.5}$ | 73.5 | 1 |
| $f$-**PLOT** | $95.4_{\pm0.2}$ | $59.6_{\pm0.9}$ | $31.3_{\pm0.4}$ | $86.5_{\pm0.1}$ | $70.1_{\pm1.3}$ | $85.8_{\pm2.2}$ | $92.4_{\pm0.2}$ | $\underline{77.5}_{\pm0.3}$ | 74.8 | 3 |
| $f$-**ProDA** | $95.4_{\pm0.2}$ | $58.6_{\pm1.0}$ | $31.3_{\pm0.8}$ | $86.6_{\pm0.1}$ | $\mathbf{70.8}_{\pm1.2}$ | $84.7_{\pm0.4}$ | $92.5_{\pm0.2}$ | $77.0_{\pm0.5}$ | 74.6 | 3 |
| $f$-**ProGrad** | $95.2_{\pm0.1}$ | $56.5_{\pm0.4}$ | $30.0_{\pm0.4}$ | $\underline{87.1}_{\pm0.1}$ | $69.3_{\pm0.1}$ | $81.3_{\pm1.4}$ | $\underline{92.6}_{\pm0.3}$ | $75.4_{\pm0.5}$ | 73.4 | 2 |
| $f$-**PromptSRC** | $94.0_{\pm0.5}$ | $58.1_{\pm0.3}$ | $31.4_{\pm0.2}$ | $86.8_{\pm0.1}$ | $70.3_{\pm0.2}$ | $85.3_{\pm0.6}$ | $92.5_{\pm0.2}$ | $75.1_{\pm0.1}$ | 74.2 | 3 |
| $f$-**KgCoOp** | $95.4_{\pm0.1}$ | $59.4_{\pm0.6}$ | $31.6_{\pm0.5}$ | $86.9_{\pm0.1}$ | $70.2_{\pm1.0}$ | $83.7_{\pm1.5}$ | $92.5_{\pm0.4}$ | $76.7_{\pm0.2}$ | 74.6 | 4 |
| $f$-**MaPLe** | $\mathbf{96.2}_{\pm0.4}$ | $\mathbf{61.2}_{\pm0.2}$ | $\underline{31.8}_{\pm0.2}$ | $\mathbf{87.6}_{\pm0.3}$ | $70.3_{\pm0.4}$ | $\mathbf{86.8}_{\pm0.1}$ | $\mathbf{92.8}_{\pm0.4}$ | $\mathbf{78.1}_{\pm0.2}$ | 75.6 | 8 |

training[16], which leverages the 16-bit floating point format to reduce GPU memory consumption and computation cost. We do not apply the AMP on DP-FPL because it requires operations such as gradient clipping in full precision format.

**Code Framework** To date, there still lacks a comprehensive and reliable evaluation of FPL algorithms for vision tasks. Zhou *et al.* [12] established the first seminal library for *centralized* prompt learning, which is later reused by a line of subsequent works. However, this library is not tailored for *federated learning*. Namely, it poses additional challenges to incorporate various federated algorithms with existing PL techniques in a flexible and scalable way.

To close this gap, we release the first framework with large-scale evaluations to push the frontier of FPL. To harvest the rapid progress from FL and PL literature, we decouple the design of the FL and PL modules, making it easier to integrate the progress from both research fields in a scalable and efficient way. We simplify and unify the interface of data-loading to achieve better adaptation of new datasets and also make it readily available for users to adapt to their customized datasets for new tasks with minimal modification. We plan to actively support more applications beyond the evaluated image classification tasks.

## D.2 Computational Resources

The experiments are conducted on a cluster consisting of multiple servers equipped with NVIDIA A100 graphic cards. We run experiments on servers equipped with the SLURM[17] job scheduler.

---

[16]https://pytorch.org/docs/stable/amp.html.

[17]https://slurm.schedmd.com/documentation.html.

Table 14: **Comparison of personal model accuracy $\alpha_{\mathrm{p}}$ (%) of FPL methods on various datasets with a ViT-B/16 image encoder.**

| Personal $\alpha_{\mathrm{p}}$ | Caltech | DTD | Aircraft | Food | Cars | Flowers | Pets | UCF | Avg. | # |
|---|---|---|---|---|---|---|---|---|---|---|
| **ZS-CLIP** | 93.5 | 45.0 | 24.3 | 85.5 | 65.6 | 68.0 | 89.2 | 67.5 | 67.3 | - |
| **PromptFL** | 95.7±0.4 | 73.3±0.6 | 43.6±0.5 | 89.1±0.6 | 76.7±1.3 | 88.5±1.0 | 92.8±1.1 | 82.5±0.6 | 80.3 | - |
| **FedOTP** | 96.2±0.5 | 75.2±2.2 | 46.8±1.1 | 90.1±0.9 | 77.7±1.8 | 91.6±0.2 | 93.1±0.7 | 84.2±1.5 | **81.9** | 8 |
| **FedTPG** | 95.7±0.2 | 73.8±0.1 | 45.2±0.3 | 88.8±0.2 | 76.2±0.8 | 89.4±0.3 | 92.3±0.5 | 83.3±0.2 | 80.6 | 4 |
| **FedPGP** | 95.2±0.3 | 73.8±0.4 | 45.6±0.4 | 88.8±0.3 | 76.8±0.7 | 87.6±0.6 | 92.6±0.4 | 82.9±0.6 | 80.3 | 4 |
| **PromptFolio** | 95.8±0.3 | 74.2±0.5 | 45.4±0.4 | 88.9±0.6 | 76.5±0.3 | 88.4±0.4 | 92.6±0.5 | 83.8±0.5 | 80.7 | 4 |
| **DP-FPL** | 95.0±0.3 | 71.2±0.5 | 42.4±0.2 | 84.2±0.8 | 75.4±0.5 | 82.8±1.6 | 92.8±0.4 | 80.0±0.6 | 78.0 | 0 |
| **$f$-CoCoOp** | 95.8±0.5 | 72.2±1.7 | 45.9±0.1 | 90.1±0.8 | 75.8±0.6 | 88.1±0.6 | 93.9±0.7 | 82.7±1.1 | 80.6 | 5 |
| **$f$-PLOT** | 95.8±0.3 | 71.5±0.3 | 44.3±1.0 | 88.7±0.4 | 76.8±1.3 | 88.8±1.8 | 92.9±1.2 | 82.0±1.2 | 80.1 | 5 |
| **$f$-ProDA** | 95.8±0.4 | 70.3±1.1 | 43.3±1.4 | 88.7±0.4 | 76.9±1.7 | 89.1±2.0 | 92.9±0.5 | 82.0±0.6 | 79.9 | 4 |
| **$f$-ProGrad** | 95.1±0.3 | 70.2±0.7 | 44.4±1.2 | 88.9±0.6 | 75.9±1.0 | 87.3±1.5 | 92.3±1.0 | 81.5±0.8 | 79.5 | 1 |
| **$f$-PromptSRC** | 94.8±0.5 | 71.2±1.4 | 44.2±0.8 | 88.4±0.8 | 76.8±0.7 | 88.0±0.9 | 92.7±1.4 | 82.4±0.8 | 79.8 | 2 |
| **$f$-KgCoOp** | 95.5±0.2 | 71.4±1.4 | 40.4±0.7 | 88.7±1.0 | 75.3±1.3 | 88.0±1.1 | 92.2±0.7 | 81.2±1.8 | 79.1 | 0 |
| **$f$-MaPLe** | 96.4±0.4 | 75.8±0.5 | 45.6±0.4 | 90.6±0.4 | 77.2±0.3 | 91.2±0.3 | 93.5±0.6 | 84.6±0.6 | **81.9** | 8 |

Table 15: **Comparison of base and novel class accuracy (%) of FPL methods with a ViT-B/16 image encoder.** Evaluation follows Table 3 except the use of a ViT-B/16 image encoder.

| Metric | Caltech | | | Aircraft | | | Cars | | | Flowers | | | Avg. | | | # |
|---|---|---|---|---|---|---|---|---|---|---|---|---|---|---|---|---|
| | $\alpha_{\mathrm{b}}$ | $\alpha_{\mathrm{n}}$ | $\alpha_{\mathrm{h}}$ | $\alpha_{\mathrm{b}}$ | $\alpha_{\mathrm{n}}$ | $\alpha_{\mathrm{h}}$ | $\alpha_{\mathrm{b}}$ | $\alpha_{\mathrm{n}}$ | $\alpha_{\mathrm{h}}$ | $\alpha_{\mathrm{b}}$ | $\alpha_{\mathrm{n}}$ | $\alpha_{\mathrm{h}}$ | $\alpha_{\mathrm{b}}$ | $\alpha_{\mathrm{n}}$ | $\alpha_{\mathrm{h}}$ | |
| **ZS-CLIP** | 95.6 | 95.5 | 95.5 | 29.5 | 34.1 | 31.6 | 67.1 | 76.5 | 71.5 | 81.6 | 68.0 | 74.2 | 68.4 | 68.5 | 68.2 | - |
| **PromptFL** | 96.6±0.3 | 95.6±0.4 | 96.1±0.3 | 32.2±1.2 | 34.6±1.0 | 33.4±1.1 | 73.2±0.8 | 74.9±0.8 | 74.0±0.3 | 87.4±0.1 | 69.1±1.1 | 77.2±0.7 | 72.4 | 68.5 | 70.2 | - |
| **FedOTP** | 97.3±0.2 | 95.4±0.8 | 96.3±0.4 | 32.5±1.8 | 34.7±2.2 | 33.5±0.5 | 72.1±0.3 | 74.5±0.1 | 73.3±0.1 | 86.5±2.5 | 70.5±0.8 | 77.6±0.6 | 72.1 | 68.8 | 70.2 | 3 |
| **FedTPG** | 96.7±0.3 | 95.7±0.6 | 96.2±0.4 | 32.8±0.4 | 33.2±0.4 | 33.0±0.2 | 73.0±0.1 | 75.9±0.3 | 74.4±0.3 | 85.8±0.4 | 68.0±0.3 | 75.9±0.3 | 72.1 | 67.8 | 69.9 | 2 |
| **$f$-CoCoOp** | 96.5±0.2 | 96.0±0.6 | 96.2±0.3 | 30.4±2.6 | 35.8±1.0 | 32.9±2.6 | 71.7±0.4 | 75.3±0.3 | 73.5±0.4 | 84.0±1.5 | 72.1±1.0 | 77.6±0.2 | 69.7 | 67.3 | 68.1 | 2 |
| **$f$-PLOT** | 96.5±0.4 | 95.7±0.6 | 96.1±0.3 | 32.7±0.8 | 34.7±0.7 | 33.7±0.8 | 71.7±0.1 | 76.2±0.1 | 73.9±0.1 | 88.4±4.3 | 68.6±2.5 | 77.2±1.2 | 54.4 | 49.8 | 51.7 | 2 |
| **$f$-ProDA** | 96.7±0.4 | 95.1±1.1 | 95.9±0.6 | 31.7±1.1 | 35.7±1.1 | 33.5±0.8 | 72.6±0.6 | 74.7±0.5 | 73.6±0.1 | 86.0±2.5 | 70.3±1.5 | 77.3±0.7 | 71.7 | 69.0 | 70.1 | 2 |
| **$f$-ProGrad** | 96.8±0.3 | 96.2±0.4 | 96.5±0.4 | 32.4±0.5 | 34.4±1.2 | 33.4±0.8 | 72.0±0.5 | 76.5±0.6 | 74.2±0.1 | 86.6±2.0 | 70.3±0.8 | 77.6±0.7 | 72.0 | 69.4 | 70.4 | 4 |
| **$f$-SRC** | 96.7±0.1 | 95.8±0.2 | 96.2±0.1 | 32.2±1.0 | 35.5±0.8 | 33.8±0.7 | 72.4±0.3 | 77.0±0.2 | 74.6±0.1 | 86.4±0.5 | 73.4±0.5 | 79.4±0.5 | 71.9 | 70.4 | 71.0 | 4 |
| **$f$-KgCoOp** | 96.7±0.6 | 96.0±0.2 | 96.3±0.3 | 33.4±0.5 | 34.3±1.0 | 33.8±0.6 | 72.9±1.0 | 75.9±0.2 | 74.3±0.5 | 88.0±2.1 | 70.6±0.4 | 78.3±0.7 | 72.8 | 69.2 | 70.7 | 4 |
| **$f$-MaPLe** | 98.4±0.3 | 97.0±0.4 | 97.6±0.3 | 34.6±0.3 | 35.9±0.2 | 35.2±0.3 | 74.2±0.5 | 77.1±0.3 | 75.6±0.4 | 88.9±0.9 | 72.7±0.8 | 80.0±0.8 | 74.0 | 70.7 | 72.1 | 4 |

## D.3 Social Impact

On the positive side, our benchmark results shed lights on the suitable application scenarios of each FPL algorithm. This allows more efficient model adaptation without centralized data collection, reducing risks of sensitive data exposure. This also promotes AI applications relying on multi-modal models in privacy-sensitive scenarios, *e.g.*, assistive technology for disabilities, federated medical imaging analysis. On the negative side, if personalized datasets in the FL network are homogenous or skewed, the personalized model via FPL algorithms may perpetuate or amplify biases (*e.g.*, cultural stereotypes) of pretrained models.

