# OpenReview forum: "FLiP: Towards Comprehensive and Reliable Evaluation of Federated Prompt Learning"
_NeurIPS.cc/2025/Datasets_and_Benchmarks_Track — NeurIPS 2025 Datasets and Benchmarks Track poster_

### Official Review · Reviewer_7aQH · 2025-06-22

**Rating:** 5
**Confidence:** 4

**Summary:**

This work introduces FLIP, a comprehensive benchmark designed to evaluate Federated Prompt Learning (FPL) algorithms for vision tasks. FPL algorithms are both practically significant and increasingly relevant, as they leverage foundation models to address key challenges in Federated Learning (FL). The proposed benchmark systematically assesses a range of recent FPL algorithms across diverse FL scenarios using 12 datasets. It evaluates efficiency, communication overhead, and performance under various data heterogeneity conditions, including unseen classes, data scarcity, and feature distribution shifts.

**Additional Feedback:**

- Overall, this is a meaningful benchmark that has the potential to contribute significantly to the advancement of the FL field. Its focus on leveraging foundation models aligns well with current trends in their widespread adoption. The work would be further strengthened by providing deeper insights and analyses.

- Since the authors state that their evaluation objectives include cross-domain distributional shifts, it would be beneficial to bring aspects related to feature shift into the main content. For example, including discussions on which components of the evaluated methods are effective under distributional shift would strengthen the paper. I recommend some papers in this field for your reference, where they discussed how to overcome domain shift, which provided valuable insights.

(i) FedBN: Federated Learning on Non-IID Features via Local Batch Normalization.

(ii) Fed-CO2: Cooperation of Online and Offline Models for Severe Data Heterogeneity in Federated Learning.

(iii) FedBiP: Heterogeneous One-Shot Federated Learning with Personalized Latent Diffusion Models.

**Dataset Code Accessibility:**

Yes

**Dataset Code Comments:**

The author provides a well-organized code base including evaluated algorithms and datasets, which is easy to follow.

**Ethical Comments:**

This paper conducts evaluation over various FPL algorithms on existing datasets. These employed datasets are public and do not have ethical issues.

**Ethical Considerations:**

No, there are no or only very minor ethics concerns

**Final Justification:**

This work presents a comprehensive benchmark for Federated Prompt Learning algorithms, aligning well with current trends in leveraging VLMs and MLLMs, where prompts play a critical role. As the authors have addressed my concerns and plan to revise the main paper to include additional experiments and discussions on feature shift in FPL, I have raised my score and recommend acceptance.

**Limitations Weaknesses:**

- Although this work shows abundant empirical results, it fails to provide deeper reflection or analyses about the results. I expect to get some insights to explain the performance difference rather than simply show the accuracy of each algorithm.

- It will make this benchmark more influential if the paper could include the discussion over the potential improvement to deficiencies of these existing algorithms. Pointing out what are the remaining challenges is important to an evaluation paper.

**Strengths Contributions:**

- This paper organizes existing FPL algorithms and conducts comprehensive evaluation experiments across a broad range of recent approaches. Such an evaluation is timely and valuable, given the increasing importance of foundation models in real-world applications.

- The evaluation framework comprehensively assesses efficiency, communication overhead, and performance under diverse federated learning scenarios, particularly those involving data heterogeneity.

- This work provides an open-source code base with unified pipelines, making it easy to follow.

---

> ### Author Rebuttal · Authors · 2025-07-31
>
> > **W1:** Additional analysis and insights about experimental results.
>
> **A1:** We sincerely appreciate the reviewer’s insightful feedback. We have presented our **key insights and discussion** in Section 5, along with the reported results for each benchmarked FL scenario. We have also summarized **key takeaways** distilled from our experimental results. Below, we provide a deeper reflection on our empirical results and offer additional insights beyond the reported accuracies.
>
> **(1) Performance Differences in Global Shared FPL**
> - *Why does f-CoCoOp underperform?* The image-conditioned prompt tuning in CoCoOp relies on stable image feature distributions, which is violated in FL due to client drift. Our hypothesis is that biased local updates cause the aggregated image-adaptive prompts to deviate from true class semantics and harm generalization.
> - *Why do regularization-based methods (f-SRC, f-KgCoOp) perform better?* These methods enforce consistency in prompt optimization across clients, mitigating client drift. This mirrors findings in FL literature (e.g., FedProx, FedProto), where regularization improves convergence under heterogeneity.
>
> **(2) Insights into Personalized FPL**
>
> The superiority of FedOTP highlights that **distribution alignment (via Optimal Transport) is highly effective for personalization**.
>
> - *Why does FedOTP outperform f-PLOT?* While both use Optimal Transport (OT), FedOTP employs imbalanced OT, which better handles the modality gap (vision vs. text) and data imbalance across clients. In contrast, f-PLOT assumes balanced alignment, leading to suboptimal adaptation.
>
> - *Why is regularization less effective for personalization?* Regularization methods (e.g., f-SRC) prioritize global consensus, which may conflict with client-specific adaptation. This suggests a trade-off: stronger regularization improves global performance but may limit personalization. Future work could explore adaptive regularization (e.g., client-specific weights).
>
>
> **(3) Base-to-Novel Generalization**
>
> The harmonic mean (αₕ) reveals that **regularization prevents overfitting to base classes, improving novel-class generalization**.
>
> - *Why does PromptFL struggle here?* Without regularization, clients overfit to local base classes. This aligns with findings in centralized prompt learning (e.g., Zhou et al., 2022), where naive prompt tuning fails to generalize.
> - *Why do knowledge-guided methods (e.g., f-KgCoOp) excel?* External knowledge from the prescribed textual features (from CLIP) provides inductive bias, helping models generalize on novel classes.
>
> **(4) Few-Shot Learning and Prompt Ensemble**
>
> The success of multi-prompt methods (FedOTP, f-ProDA, f-SRC) suggests that ensemble mitigates few-shot bias. **Diversity in prompts reduces sample selection bias from limited data**. This parallels findings in meta-learning, where multi-model aggregation improves few-shot performance.
>
> **(5) Catastrophic Forgetting in Client Sub-Sampling**
>
> The advantage of regularization-based methods under partial participation (Table 5) suggests that **consistent optimization objectives reduce forgetting**. This implies the mechanism that regularization (e.g., f-SRC’s textual consistency loss) acts as a stabilizer, preventing abrupt parameter shifts when clients join/leave.
>
> **(6) Cost-Performance Trade-offs**
>
> The inefficiency of f-CoCoOp (despite extra parameters) indicates that **FPL favors simplicity**. Unlike centralized settings, where additional capacity helps, FL’s communication bottlenecks and client drift make lightweight, robust methods more practical.
>
> **(7) High-level intuition of the effectiveness of PromptFL**
>
> Beyond these insights summarized for each scenario, we also present the high-level intuition of the strong competitive performance of the simple baseline (PromptFL).
> First, train-from-scratch FL methods usually involve millions of learnable parameters (e.g., 11.7M for ResNet18). These parameters are prone to conflict with each other during global aggregation when learned from skewed local data. In contrast, **federated prompt learning only tunes a tiny set of parameters** (e.g., 2K when prompt length is 4), and it is less likely to cause conflicting updates.
>
> Second, existing work demonstrates that in train-from-scratch FL, the classification head (last linear layer) has the lowest feature similarity (highest bias). However, in FPL, **learnable prompt parameters generally reside in the early layer of the textual or image encoder**. Therefore, prompts are less likely to be impacted by data heterogeneity.
>
> Finally, an existing work[2] also suggests that **pre-training helps align client updates** by reducing gradient diversity among clients. This also partially explains the competitiveness of PromptFL under the global-shared FPL scenario.
>
>
> > **W2:** Discussion over deficiencies, potential improvement, and remaining challenges.
>
> **A2:** Below, we explicitly outline the deficiencies of existing FPL algorithms identified through our benchmark, along with actionable improvement directions. We also elaborate on remaining challenges to guide future research.
>
> (1) Most methods excel only in either global or personalized settings. For example, regularization-based methods (f-SRC) sacrifice personalization for global consistency. A similar dilemma also happens in base-to-novel class generalization. These dilemmas motivate further research on hybrid approaches (e.g., adaptive regularization strength per client) or dynamic weighting schemes to balance the trade-off.
>
> (2) Few-shot generalization shows clear degradation in low-shot settings. While expected, reliance on ensemble methods increases computational costs for local optimization, which cannot be ignored, given the high computational overhead of foundation models. This leaves room for further improvement.
>
> (3) The performance under feature shift data heterogeneity (Table 9) and cross-domain generalization (Table 10) remains limited, underscoring the challenges of FPL against feature shift data heterogeneity. For example, even when initialized by the large foundation models, all FPL methods fail to generalize on the Quickdraw dataset. This necessitates future research to enhance the resilience of FPL methods under domain shifts.
>
>
> > **W3:** More discussion on feature shift to address cross-domain distributional shifts under FPL.
>
> **A3:** We thank the reviewer for these valuable suggestions! Indeed, we are particularly interested in addressing the feature shift problem of FPL. Following your suggestions, we will rearrange the main text to include additional experiments and discussion on feature shift of FPL.
>
> Regarding your recommended works, [3] designs local Batch Normalization (BN) layers to reduce the impact of feature shift. [4] incorporates an online and offline model, integrating generic knowledge and specialized domain knowledge to alleviate feature shift. [5] personalizes a pretrained diffusion model to synthesize images with domain properties. While these methods offer insights for addressing feature shift in FL, they work in very different ways compared with federated prompt learning.
>
> As discussed above, FPL typically tunes prompt embeddings while keeping the pretrained foundation models fixed. Therefore, the resilience against feature shift is highly related to the way of optimizing prompt embeddings. In contrast, the referred works generally optimize the entire model or resort to additional trainable models.
>
> A recent approach counteracts the feature shift in FPL by designing *domain-aware prompts* (e.g., "A photo of a [class] with the domain of [domain]") to capture domain-specific knowledge [6]. Unfortunately, the code of this work was not publicly available as of NeurIPS2025's rebuttal. We will reproduce this work and investigate further how to reduce feature shift in FPL.
>
> Notably, based on our benchmark results, distilling knowledge from a handcrafted domain-aware prompt could be an effective way, given the success of regularization-based methods such as PromptSRC [7] under FPL. In addition, motivated by [8,9], our ongoing work also explored the channel importance of multi-modal models under domain-mixed FPL training. Although these explorations are beyond the scope of this benchmark, they provide promising hints to address feature shift problem via adjusting channel importance scores.
>
> **References**
>
> [1] No Fear of Heterogeneity Classifier Calibration for Federated Learning with Non-IID Data, NeurIPS2021
>
> [2] Where to Begin on the Impact of Pre Training and Initialization in Federated Learning, ICLR2023
>
> [3] FedBN: Federated Learning on Non-IID Features via Local Batch Normalization, ICLR2021
>
> [4] Fed-CO2: Cooperation of Online and Offline Models for Severe Data Heterogeneity in Federated Learning, NeurIPS2023
>
> [5] FedBiP: Heterogeneous One-Shot Federated Learning with Personalized Latent Diffusion Models, CVPR2025
>
> [6] DiPrompT Disentangled Prompt Tuning for Multiple Latent Domain Generalization in Federated Learning, CVPR2024
>
> [7] Self-regulating Prompts Foundational Model Adaptation without Forgetting, ICCV2023
>
> [8] DePT Decoupled Prompt Tuning, CVPR2024
>
> [9] Channel Importance Matters in Few-Shot Image Classification, ICML2022

---

> > ### Comment · Reviewer_7aQH · 2025-08-05
> >
> > Thank you for your response, which addressed my concerns. As the authors plan to revise the main text to include additional experiments and discussions on the feature shift in FPL, I am willing to raise my score.

---

### Official Review · Reviewer_8uAC · 2025-06-29

**Rating:** 5
**Confidence:** 4

**Summary:**

This work seeks to provide a comprehensive and reliable evaluation of federated prompt learning algorithms. To achieve this, the authors introduce FLiP, a benchmark with open-sourced code that rigorously evaluates FPL methods across a wide spectrum of FL training protocols, scenarios, datasets and evaluation metrics. This benchmark sheds light on several important questions for this topic, such as personalized accuracy, cost-effectiveness, and performance under challenging heterogeneity and data scarcity. The experimental results underscore the shortcomings of the prevailing assessment protocols used in existing FPL works. The authors also articulated the reason of effectiveness of FPL methods under different scenarios and reveal challenges to be addressed to motivate further research.

**Dataset Code Accessibility:**

Yes

**Ethical Considerations:**

No, there are no or only very minor ethics concerns

**Final Justification:**

After reading the author's response, I prefer to increase my score.

**Limitations Weaknesses:**

(1) The rationale behind the hyperparameter choice for the FPL protocols described in Section 4.3 is not sufficiently discussed.
(2) How is hyperparameter tuning performed for each FPL method?

**Strengths Contributions:**

Existing works on FPL generally consider simple scenarios and baselines for comparison. This work contains impactful contributions by going beyond conventional FPL evaluations, including more comprehensive benchmark criteria across diverse datasets and a set of scenarios. Experiments are also very extensive with repeated trials that report statistical errors, and the evaluation metrics and scenarios are also well-justified for the FPL tasks. The necessity to develop such a new benchmark and its distinctive features compared to existing benchmarks, such as Profit is clearly justified in the related works. The code is well-documented with detailed instructions. Limitations regarding safety and robustness of federated prompt learning methods are discussed by authors.

The results contain notable insights. For instance, it shows that centralized prompt learning methods can be competitive against existing FPL counterparts. Key takeaways are summarized for different scenarios, highlighting the specific pros and cons of the FPL methods, which provides useful guidance to deploy FPL more effectively.

The writing is easy to follow, with clear explanations of benchmark design, implementation details, and experimental settings. I also appreciate its benchmark design with modularized components, which makes the code extensible and easy to modify.

---

> ### Author Rebuttal · Authors · 2025-07-31
>
> > **W1:** The rationale behind the hyperparameter choice for the FPL protocols described in Section 4.3 is not sufficiently discussed.
>
> **A1:** Fine-tuning pretrained models on downstream tasks generally relies on small learning rates. Following established practices in (federated) prompt learning literature [1-3], we adopt a relatively small initial learning rate (0.002) to ensure stable optimization and strong empirical performance. Moreover, we implement a cosine learning rate scheduler to guarantee sufficient convergence under federated training.
>
> In terms of the communication round, prior work [1,2] demonstrates that FPL typically converges within 30-50 communication rounds. Therefore, we set 50 rounds in each experiment. This efficiency arises because FPL leverages rich pretrained features rather than training from scratch, significantly reducing the required communication rounds compared to conventional federated learning approaches.
>
> > **Q2:** How is hyperparameter tuning performed for each FPL method?
>
> **A2:** We have summarized the details of hyperparameter tuning for each method in Table 7 (Appendix B). We use grid search to tune hyperparameters for each algorithm with a set of variants similar to those reported in original papers.
>
> **Reference**
>
> [1] Guo et al., pFedPrompt: Learning Personalized Prompt for Vision-Language Models in Federated Learning, WWW 2023.
>
> [2] Li et al., Global and Local Prompts Cooperation via Optimal Transport for Federated Learning, CVPR 2024.
>
> [3] Zhou et al., Conditional Prompt Learning for Vision-Language Models, CVPR 2022.

---

> ### Comment · Reviewer_8uAC · 2025-08-04
>
> After reading the author's response, I prefer to increase my score.

---

### Official Review · Reviewer_TVqA · 2025-07-01

**Rating:** 5
**Confidence:** 3

**Summary:**

This paper presents FLIP, a comprehensive benchmark for evaluating Federated Prompt Learning methods.

**Dataset Code Accessibility:**

NA; not applicable to this submission (e.g., no new dataset, benchmark, code, or data provided)

**Ethical Considerations:**

No, there are no or only very minor ethics concerns

**Final Justification:**

I keep the positive for this paper.

**Limitations Weaknesses:**

1) Why do regularization-based methods (f-SRC, f-KgCoOp) outperform others in client sub-sampling (Table 5)?
2) Table 3 (base/novel generalization) lacks comparison to non-FL baselines.

**Strengths Contributions:**

This paper evaluates 13 algorithms across 3 FL protocols, 12 datasets, and 6 distinct scenarios. Provides standardized metrics (e.g., superiority indicator, harmonic mean for base/novel classes) and open-source code, ensuring reproducibility.

---

> ### Author Rebuttal · Authors · 2025-07-31
>
> > **Q1:** Why do regularization-based methods (f-SRC, f-KgCoOp) outperform others in client sub-sampling (Table 5)?
>
> **A1:** Client sub-sampling in FL causes continuously shifting data distribution in each communication round due to clients' temporarily joining and quitting. This results in a detrimental forgetting effect on the previously learned data. Regularization-based methods introduce an additional common optimization objective among clients, which alleviates the forgetting effect.
>
> > **W2:** Table 3 (base/novel generalization) lacks comparison to non-FL baselines.
>
> **A2:** In this work, we aim to benchmark **federated** prompt learning approaches. Therefore, the non-FL baselines are adapted to our benchmark by integrating them with the FedAvg method. This ensures consistent experimental settings and fair comparison.

---

### Official Review · Reviewer_5cq6 · 2025-07-02

**Rating:** 4
**Confidence:** 4

**Summary:**

This paper addresses the lack of a standardized evaluation framework for Federated Prompt Learning (FPL). To this end, it proposes FLIP, a comprehensive and reliable benchmark for FPL. FLIP systematically evaluates 13 leading centralized and federated prompt learning algorithms on 12 datasets, under 3 federated protocols and across 6 challenging scenarios, including data heterogeneity, few-shot learning, and cross-domain generalization. Through extensive experiments, this work provides valuable insights into the performance, generalization, and cost-effectiveness of various FPL methods. It also releases a modular, open-source codebase to promote fairness and reproducibility in future research in this field.

**Dataset Code Accessibility:**

Yes

**Ethical Considerations:**

No, there are no or only very minor ethics concerns

**Final Justification:**

The authors provided a clear and detailed clarification of potential flaws of the submission. Additional experiments regarding the effectivity of the FedAvg for adaptation are conducted by replacing the basic FedAvg with FedNova and FedOpt, and the quantitative results reported in the rebuttal indicate the effectiveness of the FedAvg adaptation. However, the reason for using FedAvg could be more sufficient if a detailed sensitivity analysis (e.g., hyper-parameters) and computational complexity analysis among FedAvg and other federated learning aggregation strategies could be presented in the submission.

Overall, most of my concerns have been addressed appropriately. Thanks for the efforts on integrating different centralized prompt learning methods with different aggregation strategies.

**Limitations Weaknesses:**

-- Inaccurate Claims and Inconsistent Details:
* The paper repeatedly claims to evaluate "13 state-of-the-art (SOTA) FPL baseline algorithms," which is misleading. According to Section 4.2, the benchmark includes only 6 native FPL algorithms, while the others are adapted from centralized methods using a basic FedAvg strategy. The effectiveness of this adaptation is an open question, making it imprecise to label them as "SOTA FPL baselines."

* The description of the experimental setup is contradictory. The main text explicitly states that all experiments were "conducted with 3 runs" in line 196, but the caption of Table 3 claims the results are based on "10 random seeds." This inconsistency undermines the reliability of that portion of the results and requires clarification.

-- Insufficient Discussion on Adapted Algorithms: While including centralized PL algorithms is a good initiative, the paper only uses the most basic FedAvg for adaptation. The paper could benefit from a deeper discussion on the limitations of this approach and how the performance of these "f-" series algorithms might change with more sophisticated aggregation strategies.

Writing Issues: In lines 151-152, Equation (2) uses some symbols (e.g., total number of classes, n) without prior definition. While not hindering comprehension, it could be more rigorous.

**Strengths Contributions:**

++ Comprehensive Benchmarking: This work represents the first large-scale benchmarking effort for FPL. It covers multiple critical evaluation dimensions, from global shared and personalized models to few-shot, novel-class, and cross-domain generalization, providing a solid foundation for a thorough understanding of FPL algorithm characteristics.

++ Promotion of Reproducible Research: The authors provide a unified, modular, and open-source FLIP codebase. This is a significant contribution to the community, as it greatly lowers the barrier for other researchers to reproduce experiments and conduct fair comparisons of new algorithms.

++ Insightful Experimental Findings: The paper does not merely present data but also distills key insights from the results. For example, identifying the strong competitive performance of a simple baseline (PromptFL) and analyzing the failure modes of directly porting centralized algorithms (like f-CoCoOp) are valuable scientific contributions that offer important guidance for future research.

---

> ### Author Rebuttal · Authors · 2025-07-31
>
> > **W1:** On the terminology of adapted centralized methods and FPL methods.
>
> **A1:** We thank the reviewer for these suggestions.
>
> Our initial intention was to convey that we evaluated 13 (federated) prompt learning algorithms. We agree that the distinction between native FPL algorithms and adapted centralized methods is important for clarity. We will explicitly differentiate between the 6 native FPL algorithms and the 7 adapted methods (using FedAvg) in Section 4.2, avoiding the blanket term "SOTA FPL baselines" in our work. The effectiveness of simple adaptation with FedAvg is discussed in our response below (**A3**).
>
>
> > **W2:** Experimental details regarding random base/novel class partition.
>
> **A2:** Table 3 presents the experimental results of base-to-novel generalization. Different splits of base and novel classes could impact FPL accuracy.
>
> For reliable evaluation, we first split the same dataset into 10 different base and novel class partitions. On each partition, we run 3 repeated experiments for each algorithm and report the average results.
>
>
> > **W3:**  Further exploration on sophisticated FL aggregation strategies.
>
> **A3:** We thank the reviewer for valuable feedback.
>
> Unlike traditional FL methods that train all parameters from scratch, FPL only tunes a small set of prompt embeddings. As a result, current FPL approaches typically don’t rely on complex aggregation strategies to handle conflicting local updates. Our benchmark results also show that a simple integration of FedAvg with existing centralized prompt learning algorithms can manifest strong competitiveness against FPL methods. We also explained the underlying reason for such competitiveness (please also refer to our response to Reviewer **7aQH**).
>
> To address your concern, we integrate centralized prompt learning methods with FedNova [1] and FedOpt [2]. FedNova applies normalized averaging while FedOpt adopts adaptive optimization methods (e.g., momentum updates) for global aggregation. Our implementation is adapted from their open-source code repository. For FedNova, we search the coefficient $\rho$ from \{ 0.1, 0.5, 0.9 \} and report the results under the best coefficient $\rho = 0.9$. For FedOpt, We set  $\beta_1 = 0.9$, $\beta_2 = 0.99$ and search the server learning rate $l_g$ from \{ 0.1, 0.2, 0.5 \}. We report results under the best $l_g = 0.2$.
>
> The results for FedNova and FedOpt are illustrated in the tables below. Despite our best efforts, we cannot obtain better results than those reported in our paper by replacing FedAvg with FedNova. For FedOpt, we highlight better results compared with FedAvg by the symbol "[]". However, these improvements are not consistent across datasets or FPL algorithms. We conclude that although these sophisticated aggregation methods could be effective, they introduce additional intricacy, presumably requiring extensive case-by-case hyperparameter tuning for optimal performance.
>
>
> |Global Acc.         |Caltech   |DTD        |Aircraft  |Food      |Cars      |Flowers   |Pets      |UCF       |Avg.  |#|
> |---|---|---|---|---|---|---|---|---|---|---|
> |**ZS-CLIP**         |86.0      |41.7       |16.6      |77.9      |55.5      |65.3      |85.7      |61.5      |61.3  |- |
> |**PromptFL**        |90.1±0.5  |54.3±2.8   |21.3±0.9  |78.3±1.5  |60.8±1.2  |81.6±1.5  |88.9±1.2  |68.1±1.3  |67.9  |- |
> |**FedOTP**          |90.2±0.8  |56.2±1.6   |20.0±0.7  |78.7±0.6  |62.0±0.5  |81.0±0.5  |88.2±0.4  |69.0±0.8  |68.2  |5 |
> |**FedTPG**          |89.2±0.4  |55.6±0.8   |18.2±1.5  |78.5±0.5  |60.7±0.6  |80.0±0.8  |88.6±0.4  |67.5±1.0  |67.2  |2 |
> |
> |***f*-CoCoOp**      |82.5±1.8  |48.9±2.5   |14.5±2.6  |75.6±2.8  |58.2±2.0  |77.0±1.4  |86.2±2.1  |62.6±2.8  |67.2  |0 |
> |***f*-PLOT**        |86.2±0.1  |56.6±1.2   |20.3±1.6  |77.6±1.4  |60.6±0.3  |82.3±0.5  |87.4±1.3  |67.2±1.1  |68.0  |2 |
> |***f*-ProDA**       |88.9±1.5  |55.3±1.0   |22.4±1.3  |78.1±0.9  |61.6±0.5  |82.5±1.2  |88.9±0.6  |66.9±2.7  |68.0  |5 |
> |***f*-ProGrad**     |78.9±1.2  |54.3±1.8   |20.8±0.8  |78.2±0.8  |60.1±0.9  |81.4±1.2  |87.6±1.2  |68.4±1.1  |66.2  |2 |
> |***f*-PromptSRC**   |90.5±0.4  |54.5±1.2   |21.2±0.8  |78.6±0.5  |61.4±0.3  |81.2±0.2  |88.0±1.0  |68.1±0.2  |68.0  |6 |
> |***f*-KgCoOp**      |90.4±0.6  |56.1±0.8   |22.6±0.9  |78.4±0.8  |61.9±0.5  |82.1±0.4  |88.6±0.8  |69.3±0.3  |68.7  |7 |
> |
>
> |Global Acc.         |Caltech   |DTD        |Aircraft    |Food        |Cars        |Flowers     |Pets      |UCF         |Avg.   |#|
> |---|---|---|---|---|---|---|---|---|---|---|
> |**ZS-CLIP**         |86.0      |41.7       |16.6        |77.9        |55.5        |65.3        |85.7      |61.5        |61.3   |- |
> |**PromptFL**        |90.4±0.4  |56.9±0.8   |22.5±0.3    |78.8±0.4    |61.8±0.4    |[84.4±0.4]  |88.9±0.6  |[70.5±0.4]  |69.3   |- |
> |**FedOTP**          |91.2±0.3  |58.5±0.9   |[22.2±0.5]  |[78.9±0.3]  |62.2±0.6    |83.2±0.6    |88.6±0.2  |69.2±0.3    |69.3   |4 |
> |**FedTPG**          |90.2±0.3  |56.2±0.6   |[19.5±0.1]  |78.8±0.6    |[60.9±0.3]  |[81.2±0.6]  |88.9±0.4  |[68.9±0.4]  |[68.1] |2 |
> |
> |***f*-CoCoOp**      |89.1±0.2  |52.0±1.5   |17.5±3.2    |74.0±1.2    |59.2±0.9    |[79.2±1.2]  |87.6±0.9  |[68.2±0.9]  |65.9   |0 |
> |***f*-PLOT**        |90.0±0.4  |[58.8±1.3] |21.2±0.6    |78.2±0.4    |[60.8±0.6]  |83.2±0.5    |88.9±0.6  |[70.2±0.7]  |68.9   |2 |
> |***f*-ProDA**       |90.1±0.5  |57.0±0.6   |22.8±0.4    |[79.3±0.2]  |61.9±0.4    |83.9±0.6    |89.0±0.3  |70.1±0.4    |69.3   |5 |
> |***f*-ProGrad**     |90.6±0.5  |[57.7±1.2] |21.6±0.3    |79.4±0.2    |[60.7±0.4]  |83.2±0.3    |88.2±0.5  |70.2±0.4    |69.0   |3 |
> |***f*-PromptSRC**   |91.4±0.4  |57.2±0.8   |21.0±0.5    |78.2±0.3    |[62.6±0.4]  |83.3±0.2    |88.4±0.2  |[70.5±0.6]  |69.1   |4 |
> |***f*-KgCoOp**      |91.2±0.2  |[58.6±0.7] |22.8±0.6    |79.2±0.3    |[61.9±0.5]  |83.6±0.4    |88.9±0.4  |[70.7±0.5]  |69.6   |7 |
> |
>
>
> > **W4:** Minor writing issues of Equation (2).
>
> **A4:** We thank the reviewer for pointing out this. We will add clear definitions of all symbols in Equation (2).
>
> **References**
>
> [1] Tackling the Objective Inconsistency Problem in Heterogeneous Federated Optimization, NeurIPS2020
>
> [2] Adaptive Federated Optimization, ICLR2021

---

> > ### Comment · Reviewer_5cq6 · 2025-08-07
> >
> > The authors provided a clear and detailed clarification of potential flaws of the submission. Additional experiments regarding the effectivity of the FedAvg for adaptation are conducted by replacing the basic FedAvg with FedNova and FedOpt, and the quantitative results reported in the rebuttal indicate the effectiveness of the FedAvg adaptation. However, the reason for using FedAvg could be more sufficient if a detailed sensitivity analysis (e.g., hyper-parameters) and computational complexity analysis among FedAvg and other federated learning aggregation strategies could be presented in the submission.
> >
> > Overall, most of my concerns have been addressed appropriately. Thanks for the efforts on integrating different centralized prompt learning methods with different aggregation strategies.

---

> > ### Author Response · Authors · 2025-08-09
> >
> > Dear Reviewer 5cq6,
> >
> > We have uploaded a new response to your suggestions regarding computational overhead and sensitivity analysis. As the discussion period is coming to a close, we would be grateful for any additional feedback you may have at your earliest convenience. Thank you again for your time and thoughtful suggestions.
> >
> > Warm regards, The authors

---

> ### Author Response · Authors · 2025-08-08
> **Further Response to Reviewer 5cq6**
>
> We sincerely appreciate your constructive feedback and your acknowledgment of our efforts to clarify and improve the manuscript. Your comments offer a crucial direction to evaluate the server-side impact (e.g., aggregation algorithms) on FPL, which has improved the rigor of our benchmark.
>
> Regarding the aggregation overhead, the computational cost of FedNova [1] on the server side is similar to FedAvg. FedOpt [2] requires the calculation of momentum updates, but it can also finish in seconds. Overall, the aggregation overhead of FedNova and FedOpt is negligible because **the size of prompt embeddings is extremely small**.
>
> Below we report experiments to reflect your suggestion on **sensitivity analysis**. To avoid the intricacy introduced by algorithm-specific hyperparameters of prompt learning methods, we choose **PromptFL** as a simple baseline, combined with different aggregation methods shown in the table below, to evaluate the hyperparameter sensitivity of different aggregation methods. We bold the best accuracy on each dataset.
>
> | Global Acc.                | Caltech      | DTD          | Aircraft     | Food         | Cars         | Flowers      | Pets         | UCF          | Avg.     |
> |----------------------------|--------------|--------------|--------------|--------------|--------------|--------------|--------------|--------------|----------|
> | **ZS-CLIP**                | 86.0         | 41.7         | 16.6         | 77.9         | 55.5         | 65.3         | 85.7         | 61.5         | 61.3     |
> | **FedAvg**                 | **91.5±0.5** | **57.6±1.3** | **22.8±0.4** | **79.2±0.1** | **62.0±0.4** | 84.0±1.7     | **89.4±0.5** | 70.1±0.8     | **69.6** |
> |                            |              |              |              |              |              |              |              |              |          |
> | **FedNova ($\rho = 0.1$)** | 90.5±0.7     | 55.3±1.6     | 21.2±1.0     | 78.0±0.8     | 58.2±1.8     | 81.0±1.2     | 88.4±1.7     | 68.2±0.7     | 67.6     |
> | **FedNova ($\rho = 0.5$)** | 90.2±0.6     | 54.7±1.2     | 20.6±0.8     | 77.8±1.2     | 58.0±0.9     | 81.7±0.8     | 88.2±1.6     | 67.8±1.0     | 67.4     |
> | **FedNova ($\rho = 0.9$)** | 90.1±0.5     | 54.3±2.8     | 21.3±0.9     | 78.3±1.5     | 60.8±1.2     | 81.6±1.5     | 88.9±1.2     | 68.1±1.3     | 67.9     |
> |                            |              |              |              |              |              |              |              |              |          |
> | **FedOpt ($l_g=0.1$)**     | 90.5±0.5     | 55.9±0.5     | 22.2±0.2     | 77.0±0.4     | 60.8±0.5     | 83.2±0.6     | 87.9±0.3     | 68.7±0.4     | 68.3     |
> | **FedOpt ($l_g=0.2$)**     | 90.4±0.4     | 56.9±0.8     | 22.5±0.3     | 78.8±0.4     | 61.8±0.4     | **84.4±0.4** | 88.9±0.6     | **70.5±0.4** | 69.3     |
> | **FedOpt ($l_g=0.5$)**     | 88.7±1.4     | 55.8±1.2     | 22.7±0.7     | 77.3±1.0     | 59.5±2.1     | 82.4±0.9     | 88.2±0.7     | 67.2±1.1     | 67.7     |
>
>
> First, the results above confirm **the stability of FedAvg** as a global aggregation method in FPL.
>
> Second, our further investigation indicates that **prompt embeddings in FPL are possibly sensitive to scaling operations**, i.e., the additional normalizing factor applied by FedNova, as evidenced by the higher standard deviations compared to other aggregation methods.
>
> We hypothesize this occurs because an improper magnitude of prompt embeddings can amplify deviations in textual feature representations, given that prompts reside in the early layers of pre-trained models and may induce cascading effects throughout the feature generation process.
>
> Finally, FedOpt’s performance is **highly dependent on the global learning rate**: an excessively small rate could impede training progress, while an overly large one may cause instability. This suggests that in FPL, adaptive aggregation methods like FedOpt require careful hyperparameter tuning. These observations also align with results in their original paper [2] (Appendix D: Experiment Hyperparameters), where an extensive grid search was explored for hyperparameter tuning.
>
> To our knowledge, whether advanced aggregation strategies can deliver **consistent improvements** over FedAvg remains an open question in FPL. Our investigation provides some empirical insights. But we also acknowledge a more complete picture requires extensive work, as such a question necessitates investigating on a different problem dimension. Beyond above results and discussion, we are also trying to integrate and evaluate Wasserstein aggregation methods [3] that leverage Optimal Transport. The results will be updated in our open-source benchmark.
>
>
> Thanks again for your insightful comments, which have been immensely helpful to our work!
>
> [1] Tackling the Objective Inconsistency Problem in Heterogeneous Federated Optimization, NeurIPS 2020
>
> [2] Adaptive Federated Optimization, ICLR2021
>
> [3] Model Fusion via Optimal Transport, NeurIPS2020

---

### Author Response · Authors · 2025-08-09
**General response**

We want to express our gratitude for the insightful comments and valuable suggestions from all reviewers. We are grateful for the recognition of our timely and important evaluation (@5cq6, @7aQH, @8uAC), comprehensive benchmark (@5cq6, @TVqA, @8uAC, @7aQH), insightful findings (@5cq6, @8uAC), and open-source and extensible code framework (@5cq6, @8uAC, @TVqA, @7aQH). The comments from all reviewers will be carefully reflected in our revised manuscript, benchmark results, and updated code repository. Below is a concise summary of all discussions for your convenience.

For Reviewer **@5cq6**:
1. **Clarification on Terminology**:
   We've explicitly differentiated native FPL methods from adapted centralized ones
   and revised the term "SOTA FPL baselines".
2. **Clarification on Base/Novel Class Partitions**:
   We've clarified that the base/novel class splits use 10 partitions, while each experiment runs 3 times.
3. **More Aggregation Strategies**:
   We've further integrated and evaluated FedNova and FedOpt in our benchmark. Our analysis reveals two key findings: (1) While minor improvements are observed in certain configurations, achieving optimal performance often requires extensive hyperparameter tuning for these aggregation strategies. (2) FedAvg demonstrates better stability, as evidenced by our hyperparameter sensitivity analysis.

For Reviewer **@TVqA**:
1. **Why do f-SRC/f-KgCoOp outperform others in client sub-sampling?**
   Client sub-sampling causes continuously shifting data distributions each round, leading to catastrophic forgetting.
   Regularization-based methods mitigate this via shared optimization objectives.

2. **Does Table 3 lack non-FL baselines?**
   No. For fair comparison, we have adapted all non-FL baselines with FedAvg in our benchmark.

For Reviewer **@8uAC**
1. **Clarification on hyperparameter choice:**
   We follow the FPL literature using: (1) a small LR (0.002) for stable optimization. (2) cosine scheduler for sufficient convergence. (3) 50 rounds due to pretrained feature efficiency.
2. **How was hyperparameter tuning done?** We apply grid-search from a set of hyperparameters adapted from original papers and hyperparameter variants.

For Reviewer **@7aQH**
1. **Experimental Insights:** We offer a detailed discussion of experimental insights:
  - Global FPL: f-CoCoOp underperforms due to client drift disrupting its image-conditioned prompts, while regularization methods (f-SRC, f-KgCoOp) excel by enforcing cross-client consistent regularization objectives.
  - Personalized FPL: FedOTP dominates via imbalanced Optimal Transport, while regularization harms personalization by over-prioritizing global consensus.
  - Base-to-Novel generalization: regularization methods improve the harmonic mean by preventing base-class overfitting.
  - Few-Shot Learning: ensemble methods (FedOTP, f-ProDA) mitigate few-shot bias.
  - Client sub-sampling: regularization-based methods reduce forgetting under partial client participation.
  - Cost-Performance trade-offs: f-CoCoOp’s inefficiency shows FPL favors simplicity.
  - Why PromptFL Works Well? (1) Low parameter conflict: fewer tunable prompts minimize update conflicts. (2)  Early-layer prompts: less affected by data heterogeneity than classification heads.  (3) Pre-training alignment: reduces gradient diversity across clients, aiding global performance.
3. **Discussion over deficiencies, potential improvement, and remaining challenges:**
  - Global vs personalization trade-off needs adaptive or hybrid approaches.
  - High computational costs of few-shot methods necessitate further exploration on efficiency.
  - Feature shift remains challenging under FPL.
4. **More discussion on Feature Shift:**
  - Current FL methods (BN layers, diffusion models) differ from FPL's training with fixed foundation models.
  - Domain-aware prompts show promise but need reproduction.
  - Knowledge distillation from domain-aware prompts and channel importance adjustment could help.


Our benchmark is under continuous development, and it will include more recent algorithms and models. We are working on integrating the recently released FAIR's Meta CLIP 2 [1] into our benchmark. Thanks again for your time and effort.

[1] Meta CLIP 2: A Worldwide Scaling Recipe. ArXiv 2025.

---

### Note · Authors · 2025-08-13

Dear ACs, SACs, and Reviewers,

We thank reviewers for their time, feedback, and engagement to strengthen our manuscript. We also thank ACs and SACs for coordinating discussions; your efforts are greatly appreciated.

Our paper presents FLiP, the first large-scale benchmark to evaluate federated prompt learning algorithms for adapting foundation models in FL. FLiP integrates 13 (federated) prompt learning algorithms within a decoupled, modular framework that offers standardized data interfaces, aligned experimental protocols, and comprehensive evaluation.  Our benchmark enables fair and reliable assessment of FPL algorithms, providing key insights of their strengths and weaknesses across diverse FL scenarios.

We refer to the general response posted below for a discussion summary. Overall, the discussion shows that our responses have addressed reviewers’ concerns and strengthened our benchmark. We thank all reviewers' **unanimous support** for our work during the discussion phase.

Warm regards,
The authors

---

### Decision · Program_Chairs · 2025-09-18

**Decision:**

Accept (poster)

**Comment:**

This paper proposed FLIP, a benchmark on federated prompt learning. Reviewers appreciate the coverage of the dataset and algorithm, while also raising several concerns on baselines, parameter tuning, and clarification. After rebuttal, there is a general consensus on acceptance with an average score of 4.75 (4, 5, 5, 5). I also appreciate the authors clearly summarizing the insights from benchmarking in the paper. That being said, the NeurIPS conference is particularly competitive this year. FL algorithms are hard to tune in practice, as suggested by reviewers, and more discussion on whether the datasets represent the practice would be appreciated.

Regardless, I would encourage the authors to incorporate reviewers' comments in improving the paper draft. For example, having a baseline on centralized setting as suggested by reviewers would help us understand more about the problem. I would also encourage the authors to keep thinking about how this benchmark can be useful for future research and the broader community.